# ON THE DESIGN AND ANALYSIS OF LLM-BASED ALGORITHMS

## ABSTRACT

We initiate a formal investigation into the design and analysis of LLM-based algorithms, i.e. algorithms that contain one or multiple calls of large language models (LLMs) as sub-routines and critically rely on the capabilities of LLMs. While LLM-based algorithms, ranging from combinations of basic LLM calls to complicated LLM-powered agent systems and compound AI systems, have achieved remarkable empirical success, the design and optimization of them have oftentimes relied on heuristics and trial-and-errors, which is largely due to a lack of formal and analytical study for these algorithms. To fill this gap, we start by identifying the computational-graph representation, task decomposition as the design principle, and some key abstractions, which then facilitate our formal analysis for the accuracy and efficiency of LLM-based algorithms, despite the black-box nature of LLMs. Through extensive analytical and empirical investigation in a series of case studies, we demonstrate that the proposed framework is broadly applicable to a wide range of scenarios and diverse patterns of LLM-based algorithms, such as parallel, hierarchical and recursive task decomposition. Our proposed framework holds promise for advancing LLM-based algorithms, by revealing the reasons behind curious empirical phenomena, guiding the choices of hyperparameters, predicting the empirical performance of algorithms, and inspiring new algorithm design. To promote further study, we include our source code in the supplementary materials.

## 1 INTRODUCTION

The rapid advancements of pre-trained large language models (LLMs) in the past few years (Bubeck et al., 2023; Wei et al., 2022a; Schaeffer et al., 2023) have given rise to the new paradigm of algorithmic problem solving by utilizing LLMs as general-purpose solvers and prompting them with task descriptions plus additional inputs that can help enhance their performance (Wei et al., 2022b; Kojima et al., 2022). Meanwhile, it is also well recognized that even the best LLMs today still exhibit various limitations, such as finite context window sizes and difficulty with complex reasoning. Some of these limitations are fundamental (Merrill & Sabharwal, 2023; Peng et al., 2024; Thomm et al., 2024; Hahn & Rofin, 2024; Wen et al., 2024; Merrill et al., 2024), and resolving them requires new breakthroughs. Moreover, in resource-constrained scenarios where using the state-of-the-art LLMs is not feasible, one might have to resort to smaller and weaker LLMs for solving complex tasks.

All these have motivated the developments of what we call *LLM-based algorithms*, namely, algorithms that contain one or multiple LLM calls as sub-routines and fundamentally rely on the capabilities of LLMs. In its most basic form, an LLM-based algorithm can be a combination of multiple LLM calls (Yao et al., 2023a; Besta et al., 2023; Wang et al., 2023). More advanced examples include LLM-powered agent systems (Mialon et al., 2023; Pezeshkpour et al., 2024; Xi et al., 2023) and compound AI systems (Zaharia et al., 2024) that augment LLMs with additional abilities like tool use and long-term memory, as well as the emerging paradigm of LLM programming (Schlag et al., 2023; Zheng et al., 2024).

LLM-based algorithms, in a similar spirit to neuro-symbolic programming (Chaudhuri et al., 2021; Gupta & Kembhavi, 2023) and learning-augmented algorithms (Mitzenmacher & Vassilvitskii, 2022; Lindermayr & Megow, 2022), combine the advantages of both LLMs and traditional algorithms. An LLM-based algorithm, designed with human's intelligence and knowledge of

algorithmic problem solving, can exhibit much better controllability and interpretability, stronger performance that is less reliant on delicate prompting or extensive trial-and-errors, and capabilities that far exceed what could possibly be achieved by directly prompting the LLM for a solution.

The rapid developments of LLM-based algorithms naturally raise the question: is it possible to provide any *formal analysis or guarantee* for LLM-based algorithms? This seems like a daunting task at first glance, due to the black-box nature of LLMs. Indeed, in prior works of this field, LLM-based algorithms have been typically designed in a heuristic manner and evaluated empirically with measurements of some error or cost metrics on certain benchmarks. In contrast, formal analysis of LLM-based algorithms, if it does exist, can bring various potential benefits, including but not limited to revealing the reasons behind curious empirical phenomena, instructing choices of hyperparameters, predicting the empirical performance of LLM-based algorithms, and even inspiring new algorithm design.

**Main contributions.** The goal of this work is to initiate a formal investigation into the design and analysis of LLM-based algorithms. Our contributions to filling this gap, and thereby advancing the field of LLM-based algorithms, are summarized as follows.

- Section 2 introduces a novel analytical framework. We start by formulating LLM-based algorithms as computational graphs, and identifying task decomposition as the design principle. We further introduce some key abstractions, based on which formal analysis for the accuracy and efficiency of LLM-based algorithms is developed.

- Through a series of case studies in Sections 3, 4 and 5, we demonstrate the proposed framework in action for various patterns of LLM-based algorithms, including parallel, hierarchical and recursive decomposition, in a wide range of diverse scenarios and tasks.

- We derive novel insights from our analytical study, which are also validated by experiments. For example, considering the hyperparameter $m$ that represents the granularity of parallel decomposition, our analysis explains why error and cost metrics of an LLM-based algorithm are monotone in $m$ in certain cases while non-monotone in others, which in turn guides the choices of $m$ for achieving the desired accuracy or efficiency. Our work exemplifies how to leverage the proposed framework for systematic design and analysis of practical LLM-based algorithms, which can potentially inspire future work in this area.

## 2 AN ANALYTICAL FRAMEWORK

This section introduces our formal framework for the design and analysis of LLM-based algorithms.

### 2.1 DEFINITION, REPRESENTATION, AND DESIGN PRINCIPLE

**Definition and computational-graph representation.** Generally speaking, an *LLM-based algorithm* is simply an algorithm that contains one or multiple LLM calls as its key components. Examples of LLM-based algorithms range from one single LLM call, to LLM-powered agent systems or compound AI systems consisting of a mixture of LLM calls and non-LLM programs.

An LLM-based algorithm can be naturally formulated as a *computational graph*. Each graph node takes some inputs from its predecessor nodes, executes certain operations, and returns some outputs. The nodes can be categorized into two types, which we refer to as LLM nodes and non-LLM nodes, demonstrated in Figure 1. Within an *LLM node*, the operations consist of a prompter that formats a prompt based on the inputs, an LLM call that processes the prompt, and a parser that extracts the targeted information from the LLM's response. The prompter and parser, designed specifically for the current node, serve as translators between natural language and traditional data structures. Within a *non-LLM node*, the operations can be anything that does not involve LLMs, e.g. a symbolic algorithm, an API call for a search engine, or a classical machine learning model.

Given such nodes, the computational graph is built by connecting them with directed edges that specify the data flows within the LLM-based algorithm. For example, Figure 2a demonstrates the pattern of parallel decomposition, which divides the input problem into parallel sub-tasks, solves each with one LLM node, and aggregates the results for the final solution; Figure 2b illustrates an algorithm for book-length summarization with chunking and incremental updating (Chang et al.,

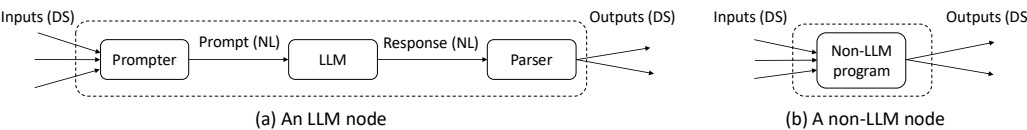

Figure 1: Two types of nodes in the computational graphs of LLM-based algorithms. We use the abbreviation "NL" for "natural language", and "DS" for "data structure".

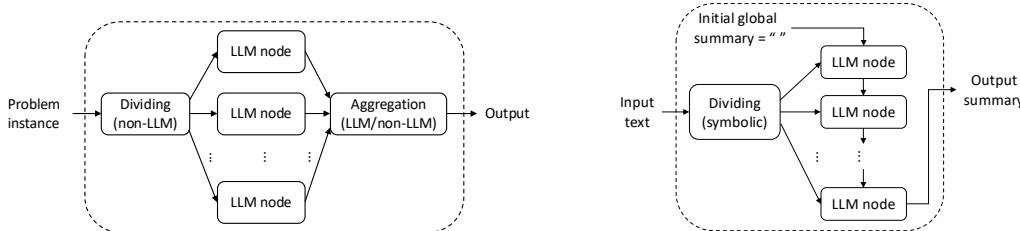

(a) Parallel decomposition. Examples of this pattern are elaborated in Section 3 and Appendix C.

(b) Book-length summarization via chunking and incremental updating (Chang et al., 2024).

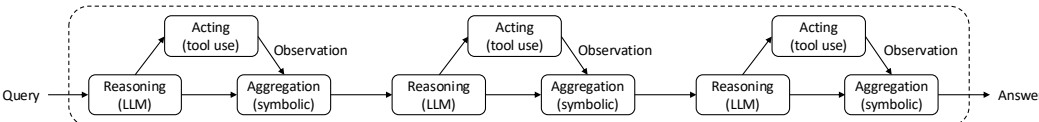

(c) The ReAct algorithm (Yao et al., 2023b). Each "acting" node represents one API call for some tool, and each "aggregation" node aggregates the outputs of its predecessor nodes, e.g. by simple concatenation.

Figure 2: Examples of computational-graph representations for LLM-based algorithms.

2024; OpenAI, 2024b); and Figure 2c demonstrates the ReAct algorithm (Yao et al., 2023b), which consists of multiple iterations of reasoning, tool use and aggregation. The computational-graph formulation is expressive enough to cover these diverse algorithms and facilitates unified analysis for all of them. The computational graph of an LLM-based algorithm can be pre-specified and static, or dynamically constructed at runtime. Configurations of an LLM-based algorithm include how task decomposition is done (reflected by the graph topology and the sub-tasks of graph nodes), the methods of prompting and parsing for each LLM node, configurations of the LLM(s) being used (e.g. the LLM nodes within one graph might use the same or different backbone LLMs), and so on.

**Design principle: task decomposition.** As shown above, an LLM-based algorithm might utilize multiple LLM calls and non-LLM programs to handle sub-tasks derived from the original problem, whose outputs together give rise to the final solution. This naturally implies *task decomposition* as a crucial principle. One key aspect of designing an LLM-based algorithm is to decompose the original task appropriately, so that each sub-task can be handled by one LLM call or non-LLM program accurately and efficiently, while the performance of the overall algorithm is also guaranteed.

Task decomposition, widely adopted in prior works on LLM-based algorithms, is beneficial and oftentimes crucial for solving problems with LLMs. For example, a large document that is well beyond the context window or long-context capability of the LLM might need to be divided into chunks, each processed by one LLM call; or, a complex reasoning process might need to be decomposed into smaller steps, handled by multiple LLM calls in coordination. It is also possible that better efficiency might be achieved with task decomposition, e.g. when the input problem can be decomposed into multiple independent sub-tasks to be solved in parallel, with a smaller end-to-end latency than autoregressive decoding within one single LLM call.[1] Despite all these benefits, it is also possible that fine-grained task decomposition can incur higher errors or costs in certain cases. Choosing the appropriate decomposition is crucial for achieving good performance of the overall algorithm, be it a trade-off between accuracy and efficiency or the best of both worlds.

---

[1]It is worth noting that one single LLM call, as a special case of LLM-based algorithms, can also be treated from the perspective of task decomposition. Thus the analysis proposed in this work still holds for this case.

**Analysis: accuracy and efficiency.** Given a task and the corresponding LLM-based algorithm, we aim to analyze its performance, namely how accurately and efficiently the algorithm solves the task, akin to analysis for any generic algorithm. This is done by analyzing for each LLM call or non-LLM program first, and then for the overall algorithm. Given that LLMs are regarded as black-box general-purpose problem solvers using natural language as input and output, we find it useful and necessary to leverage certain abstractions, to be introduced in Section 2.2, in order to facilitate formal analysis that will be presented in Section 2.3. One practical usage of formal analysis is to predict the performance of an LLM-based algorithm before actually running it, which will in turn help to optimize certain hyperparameters in the configurations or inspire better algorithm design.

## 2.2 KEY ABSTRACTIONS

We introduce a few key abstractions, summarized in Figure 3, that will facilitate our formal analysis.

**Error and cost metrics.** The accuracy and efficiency of each graph node and the overall LLM-based algorithm can be quantified by certain error and cost metrics respectively. The performance of individual graph nodes, together with the graph topology, implies the performance of the overall algorithm. For each specific task or algorithm, one might define multiple error metrics and cost metrics, which can be analyzed in a unified manner within our proposed framework.

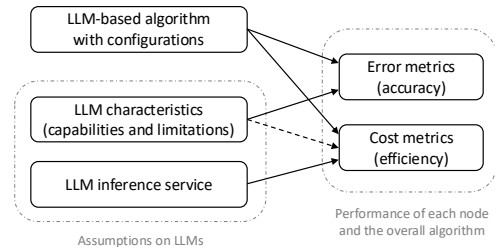

Figure 3: Key abstractions and their relation.

Error metrics are task-specific in general, and graph nodes within the same algorithm might have different error metrics.[2] Cost metrics, on the other hand, are largely task-agnostic. Examples include the total length of prompts and generated texts within one run of the algorithm, which are especially relevant to the financial costs when the algorithm uses a proprietary LLM accessed via commercial API calls. Another example is the end-to-end latency, namely the time complexity, which can be impacted by parallelism of LLM calls, an important aspect of LLM inference service in practice. Other possible cost metrics include the peak memory usage, the total number of LLM calls, FLOPs, energy consumption, carbon emission, and so on. Unless specified otherwise, we focus on the costs of the LLM nodes and neglect those of the non-LLM nodes, since the latter is much smaller than the former in all concrete scenarios that we will consider later in this work.

**LLM characteristics and inference service.** Characteristics of LLMs, namely their *capabilities and limitations*, determine what the generated text will be for a specific prompt, and thus directly affect the error metrics of graph nodes and the overall algorithm. They also affect the cost metrics *indirectly*, via the lengths of prompts and generated texts. Assumptions on LLM characteristics can be task-specific or task-agnostic. Assumptions on LLM inference service, on the other hand, are task-agnostic and only affect the cost metrics, not error metrics. They determine how cost metrics are dependent on the lengths of prompt and generated text for each LLM call, the parallelism of multiple LLM calls, among other factors. While LLM inference service in practice can be very diverse, we will see that unified and formal analysis is possible with appropriate abstractions of it.

## 2.3 FORMAL ANALYSIS: ACCURACY AND EFFICIENCY

Given the above formulations and abstractions, we are ready to conduct formal analysis for the accuracy and efficiency of LLM-based algorithms. Our approach is to first analyze the error and cost metrics for each individual node within the computational graph; these, combined with the graph topology, lead to results about the error and cost metrics of the overall algorithm. Unless otherwise specified, our analysis is deterministic, for a given task instance and fixed random seed(s).

---

[2]Throughout this work, we use "accuracy" to refer to the broader concept of "quality", and an "error metric" can be any metric that measures how much the output of an algorithm deviates from certain criteria.

**Analysis of error metrics.** The error metrics of the output of each LLM node, calculated with respect to what the output should have been if all nodes accomplish their tasks with exact accuracy, depend on the characteristics (i.e. capabilities and limitations) of the LLM, as well as the specific problem instance and random seed(s). For each node $v$, the error of its output $y$ can be bounded by some function $f_v$ of the errors of its inputs $x_1, x_2, \ldots, x_k$, i.e. the outputs of its predecessor nodes:

$$\mathcal{E}(y) \leq f_v\big(\mathcal{E}(x_1), \mathcal{E}(x_2), \ldots, \mathcal{E}(x_k)\big). \tag{1}$$

The function $f_v$ can be a linear function in a counting task to be introduced in Section 3.2, or the minimum operator in code generation where multiple code samples are generated and the best one that passes all or the most test cases is chosen. More examples of $f_v$ will be demonstrated in our case studies later in this work. Finally, the error metrics of the overall algorithm are exactly those of the particular graph node that generates the final solution returned by the algorithm.

**Analysis of cost metrics.** Let us first consider the cost of one LLM call, which consists of a *prefilling phase* with a prompt of length $\mathcal{L}_{\text{pre}}$, and a *decoding phase* that generates text of length $\mathcal{L}_{\text{dec}}$. In our framework, it is assumed that the cost $\mathcal{C}$ of one LLM call can be bounded by

$$\mathcal{C} \leq \mathcal{C}(\text{prefilling}) + \mathcal{C}(\text{decoding}) = \mathcal{C}_{\text{pre}}(\mathcal{L}_{\text{pre}}) + \mathcal{C}_{\text{dec}}(\mathcal{L}_{\text{pre}}, \mathcal{L}_{\text{dec}}) =: \mathcal{C}_{\text{LLM}}(\mathcal{L}_{\text{pre}}, \mathcal{L}_{\text{dec}}). \tag{2}$$

The functions $\mathcal{C}_{\text{pre}}$, $\mathcal{C}_{\text{dec}}$ and $\mathcal{C}_{\text{LLM}}$ are specific to each LLM call, and depend on the choices of cost metrics and LLM inference service. For example, they might be linear functions when the financial costs of LLM API calls charged by tokens are under consideration, or quadratic functions for the latency of inference by Transformers with full attention (Vaswani et al., 2017).

Next, we consider the cost of the overall algorithm, which is a function of the costs of all LLM nodes within its computational graph. This function might be a simple sum (e.g. for LLM API costs) or something more complex. One particular case of interest is the latency in a setting of *ideal parallelism* with maximum degree $p$, which assumes that the inference service is capable of processing $k \leq p$ independent LLM calls in parallel within time $\max_{i \in [k]} \mathcal{C}_i$, where $\mathcal{C}_i$ denotes the latency of processing the $i$-th LLM call alone. If $k > p$ instead, a natural idea is to divide them into $g = \lceil k/p \rceil$ groups, with each group of $p$ LLM calls processed in parallel; the end-to-end latency $\mathcal{C}$ in this case is thus

$$\mathcal{C} = \sum_{j \in [g]} \max \Big\{ \mathcal{C}_{(j-1)p+1}, \mathcal{C}_{(j-1)p+2}, \ldots, \mathcal{C}_{\min\{jp, k\}} \Big\}. \tag{3}$$

This setting of ideal parallelism will be considered throughout our case studies in Sections 3 and 4,

**Case studies.** We have presented our unified and systematic analysis for generic LLM-based algorithms. Through a series of case studies to be presented in Sections 3, 4 and 5, we will demonstrate the proposed framework in action and derive novel insights for diverse patterns of LLM-based algorithms, including parallel, hierarchical and recursive decomposition.

## 3 PARALLEL DECOMPOSITION

This section focuses on *parallel decomposition*, a basic MapReduce-like (Dean & Ghemawat, 2008) pattern visualized in Figure 2a. An algorithm of this pattern divides the input problem into multiple independent sub-tasks, solves each with one LLM node, and aggregates the results with an LLM or non-LLM node for the final solution. The intermediate sub-tasks can be solved sequentially or in parallel, which has impacts on certain cost metrics. This basic pattern of task decomposition can be used as a building block for more sophisticated algorithms. Despite its simplicity, interesting analysis and a wide variety of concrete tasks and algorithms can be derived from it.

In the following, Section 3.1 introduces our general analysis for parallel decomposition, while Sections 3.2 and 3.3 further demonstrate two concrete examples, namely counting and retrieval. Analysis with the proposed framework for more concrete examples (including sorting, retrieval-augmented generation, and long-text summarization) can be found in Appendix C.

### 3.1 NOTATIONS AND ANALYSIS

We define some formal notations that will be useful in our analysis. Let $n$ denote the size of the input problem instance, e.g. the number of tokens in a piece of text or the length of a list. Let $k$ denote the

number of parallel sub-tasks after decomposition, and $m_i$ denote the size of the $i$-th sub-task, where $i \in [k]$. It is assumed that $m_i \leq \overline{m}$ for some maximum value $\overline{m}$ that can fit into the context window of the LLM. For most of our analysis, we will assume for simplicity that $m_i = m$ for all $i \in [k]$, and that $k = O(n/m)$. We denote $p \geq 1$ as the maximum degree of ideal parallelism supported by LLM inference service. Finally, let $\mathcal{L}_{\sf sys}$ be an upper bound for the length of the system prompt of each LLM call, which includes everything in the prompt except for the size-$n$ input problem instance. The presence of $\mathcal{L}_{\sf sys}$, which can be large in practice, is essentially due to the fact that the LLM is used as a general-purpose sub-routine, hence specifying the concrete task for each LLM call constitutes part of the complexity.

For notational convenience, we will often write $f(n) \lesssim g(n)$ in place of $f(n) = O(g(n))$, which means there exists a universal constant $C > 0$ such that $f(n) \leq C \cdot g(n)$ for any positive integer $n$. In addition, $f(n) \asymp g(n)$ means $f(n) \lesssim g(n)$ and $g(n) \lesssim f(n)$ both hold. To further simplify notation, we will often omit the big-O notation in the input arguments of cost functions $\mathcal{C}_{\sf pre}$, $\mathcal{C}_{\sf dec}$ and $\mathcal{C}_{\sf LLM}$; for example, given a prompt of length $\mathcal{L}_{\sf sys} + O(n)$, we will write the cost of the prefilling phase as $\mathcal{C}_{\sf pre}(\mathcal{L}_{\sf sys} + n)$ rather than $\mathcal{C}_{\sf pre}(\mathcal{L}_{\sf sys} + O(n))$.

**Analysis of cost metrics.**   We assume for simplicity that all parallel LLM nodes share the same LLM model and inference service, and thus the same cost functions $\mathcal{C}_{\sf pre}$, $\mathcal{C}_{\sf dec}$ and $\mathcal{C}_{\sf LLM}$.

In many cases, the cost of the overall algorithm is a simple sum of the costs of all LLM calls, e.g. the financial cost for commercial LLM API calls, or the end-to-end latency when all LLM calls are executed sequentially. In such cases, we can write the total cost as $\mathcal{C} = \mathcal{C}(\text{sub-tasks}) + \mathcal{C}(\text{aggregation})$, where $\mathcal{C}(\text{aggregation})$ is the cost of the final aggregation step, and

$$\mathcal{C}(\text{sub-tasks}) = k \times \mathcal{C}(\text{one sub-task}) \leq k \times \mathcal{C}_{\sf LLM}(\mathcal{L}_{\sf sys} + m, \mathcal{L}_{\sf dec}) \lesssim n \times \frac{\mathcal{C}_{\sf LLM}(\mathcal{L}_{\sf sys} + m, \mathcal{L}_{\sf dec})}{m}. \tag{4}$$

Here, the first inequality follows Eq. (2), the second follows $k \lesssim n/m$, and $\mathcal{L}_{\sf dec} = \mathcal{L}_{\sf dec}(m)$ is task-specific, which can be $O(1)$ or $O(m)$ for example. This basic analysis already provides some hints for tuning the hyperparameter $m$ from the perspective of minimizing costs. To see this, let us assume for simplicity that $\mathcal{L}_{\sf dec} = O(1)$. If $\mathcal{C}_{\sf LLM}(\mathcal{L}_{\sf sys} + m, 1)$ grows with $m$ at a linear or sub-linear rate, then the right-hand side of Eq. (4) is monotonely decreasing in $m$, which means $\mathcal{C}(\text{sub-tasks})$ is minimized at $m = \min\{n, \overline{m}\}$. On the other hand, it is well known that a Transformer model with full attention suffers from quadratic complexity in long-sequence processing. A general assumption would be $\mathcal{C}_{\sf LLM}(\mathcal{L}_{\sf sys} + m, 1) \leq \alpha \times (\mathcal{L}_{\sf sys} + m)^2 + \beta \times (\mathcal{L}_{\sf sys} + m) + \gamma$, which takes into account a more precise characterization of the FLOPs, as well as memory IO for loading model weights and KV caches (Agarwal et al., 2023). In this case, we have

$$\mathcal{C}(\text{sub-tasks}) \lesssim n \times \frac{\mathcal{C}_{\sf LLM}(\mathcal{L}_{\sf sys} + m, 1)}{m} \leq n \times \frac{\alpha \times (\mathcal{L}_{\sf sys} + m)^2 + \beta \times (\mathcal{L}_{\sf sys} + m) + \gamma}{m}$$

$$= n \times \left( \alpha \times m + \frac{\alpha \times \mathcal{L}_{\sf sys}^2 + \beta \times \mathcal{L}_{\sf sys} + \gamma}{m} + 2 \times \alpha \times \mathcal{L}_{\sf sys} + \beta \right),$$

and the right-hand side is minimized at $m = \sqrt{\mathcal{L}_{\sf sys}^2 + \mathcal{L}_{\sf sys} \times \beta/\alpha + \gamma/\alpha}$.

The above analysis can be extended to the case with parallelism, which is especially relevant to the end-to-end latency of the overall algorithm. Considering parallel LLM calls for homogeneous sub-tasks in the setting of ideal parallelism introduced earlier, we have

$$\mathcal{C} = \mathcal{C}(\text{sub-tasks}) + \mathcal{C}(\text{aggregation}), \quad \text{where} \quad \mathcal{C}(\text{sub-tasks}) \leq \left\lceil \frac{k}{p} \right\rceil \times \mathcal{C}_{\sf LLM}(\mathcal{L}_{\sf sys} + m, \mathcal{L}_{\sf dec})$$

according to Eq. (3). For large $m$ such that $k \leq p$, we have $\lceil k/p \rceil = 1$ and thus $\mathcal{C}(\text{sub-tasks}) \leq \mathcal{C}_{\sf LLM}(\mathcal{L}_{\sf sys} + m, \mathcal{L}_{\sf dec})$ is monotonely *increasing* in $m$. On the other hand, for sufficiently small $m$ and large $k$, we have $\lceil k/p \rceil \approx k/p$, and thus the upper bound for $\mathcal{C}(\text{sub-tasks})$ can be approximated by $\mathcal{C}(\text{sub-tasks}) \leq (k/p) \times \mathcal{C}_{\sf LLM}(\mathcal{L}_{\sf sys} + m, \mathcal{L}_{\sf dec}) \lesssim (n/p) \times \mathcal{C}_{\sf LLM}(\mathcal{L}_{\sf sys} + m, \mathcal{L}_{\sf dec})/m$, which might be monotonely *decreasing* in $m$ if $\mathcal{C}_{\sf LLM}(\mathcal{L}_{\sf sys} + m, \mathcal{L}_{\sf dec})$ grows with $m$ at a (sub-)linear rate. In this case, the overall cost $\mathcal{C}(\text{sub-tasks})$ is minimized by $k \asymp p$ and hence $m \asymp n/p$.

One implication of the above analysis is, the optimal value of $m$ that minimizes costs might depend on the choices of cost metrics and assumptions of LLM inference service, among other factors.

**Analysis of error metrics.** As explained earlier in Section 2.2, error metrics are mostly task-specific, and thus the analysis of error metrics need to be done in a case-by-case manner. For a given task, error metrics for each homogeneous and parallel sub-task typically increases with the size $m$, although this is not always the case. Error metrics of the overall algorithm after the final aggregation step also depend on the specific task, the method of aggregation, and choices of metrics. We will soon demonstrate the analysis of error metrics in several concrete examples.

### 3.2 EXAMPLE: COUNTING

As a warm-up exercise, we consider a simple counting task formulated as follows: given a string of length $n$ consisting of letters and digits, the task is to count the number of digits in it. This can be seen as an abstraction or synthetic version of more generic counting tasks in practice.

**Algorithm.** We consider the following algorithm with the pattern of parallel decomposition. It first divides the input string into $k$ disjoint sub-strings of lengths $m_1, m_2, \ldots, m_k$. Then, for each $i \in [k]$, one LLM call is invoked to count the number of digits in the $i$-th sub-string, whose answer is denoted by $y_i$. The final solution returned by the algorithm is then $y = \sum_{i \in [k]} y_i$. For each LLM call, we prompt the LLM to generate its answer directly without intermediate steps, thus it is reasonable to assume that the text generated by each LLM call has length $\mathcal{L}_{\mathsf{dec}} = O(1)$.

**Analysis.** Let us assume for notational convenience that $m_i = m$ for all $i \in [k]$, and $k = n/m$ is an integer. We first consider the accuracy of this algorithm. Denote by $y^\star$ and $y_i^\star$ the ground-truth count for the complete string and the $i$-th sub-string, respectively. If $\mathcal{E}$ represents the absolute counting error, then $\mathcal{E}(y) := |y - y^\star| = |\sum_{i \in [k]} (y_i - y_i^\star)| \leq \sum_{i \in [k]} |y_i - y_i^\star| = \sum_{i \in [k]} \mathcal{E}(y_i)$. If we let $\mathcal{E}$ represent the normalized counting error instead, then

$$\mathcal{E}(y) := \frac{|y - y^\star|}{n} \leq \frac{1}{n} \sum_{i \in [k]} |y_i - y_i^\star| = \frac{1}{k} \sum_{i \in [k]} \frac{|y_i - y_i^\star|}{m} = \frac{1}{k} \sum_{i \in [k]} \mathcal{E}(y_i).$$

Since a smaller value of $m$ makes each sub-task easier, it is reasonable to expect that the overall error $\mathcal{E}(y)$ with this metric will also become smaller as $m$ decreases.

Next, our analysis of efficiency follows Section 3.1. Here, we have $\mathcal{L}_{\mathsf{dec}} = O(1)$ by assumption, and $\mathcal{C}(\mathsf{aggregation}) = 0$ since the final aggregation step is done by a non-LLM node. Considering the total prefilling length and decoding length as cost metrics, one has $\mathcal{C}(\mathsf{prefilling}) \lesssim k \times (\mathcal{L}_{\mathsf{sys}} + m) = n \times (\mathcal{L}_{\mathsf{sys}}/m + 1)$ and $\mathcal{C}(\mathsf{decoding}) \lesssim k \times 1 = n/m$, both of which are monotonely decreasing in $m$. More generally, the sum of costs of all LLM calls is $\mathcal{C} = \mathcal{C}(\mathsf{sub\text{-}tasks}) \leq k \times \mathcal{C}_{\mathsf{LLM}}(\mathcal{L}_{\mathsf{sys}} + m, 1) = n \times \mathcal{C}_{\mathsf{LLM}}(\mathcal{L}_{\mathsf{sys}} + m, 1)/m$. The optimal choice of $m$ under various conditions has been discussed in Section 3.1. Finally, considering the end-to-end latency with parallelism degree $p$, we have $\mathcal{C} = \mathcal{C}(\mathsf{sub\text{-}tasks}) \leq \lceil k/p \rceil \times \mathcal{C}_{\mathsf{LLM}}(\mathcal{L}_{\mathsf{sys}} + m, 1)$, which is minimized around $k = p$ and $m = n/p$ under the assumptions explained in Section 3.1.

In sum, the above analysis characterizes the accuracy and efficiency of the counting algorithm, with a particular focus on how they are impacted by the hyperparameter $m$ indicating the granularity of parallel decomposition. Empirical validation of our analysis can be found in Appendix C.1.

### 3.3 EXAMPLE: RETRIEVAL

For another application of parallel decomposition, let us consider the task of question answering that requires retrieving some key information from a long piece of text, akin to the needle-in-a-haystack benchmark (Kamradt, 2023). For example, suppose that a key message (the needle) of the form "The passcode to the {targeted object, e.g. red door} is {6-digit passcode}" is randomly inserted into a piece of long text (the haystack), and the algorithm is asked to answer "What is the passcode to the {targeted object}?". To make this task more challenging and fun, we further assume that the haystack consist of alike sentences of the form "The passcode to the {colored object} is {6-digit passcode}", with colored objects different from the targeted object. This allows us to investigate both sides of retrieval capabilities of LLMs and LLM-based algorithms: retrieving the targeted information correctly, while avoiding being confused or misled by background information that might seem relevant to the question (Shi et al., 2023). In the following, we highlight our algorithm design and analysis of accuracy, with full details of this example deferred to Appendix C.3.

We consider the algorithm below that follows the pattern of parallel decomposition. It first divides the input text of length $n$ into $k$ chunks of lengths $m_1, \ldots, m_k$. Then for each chunk, one LLM call is invoked to try to answer the question based on that chunk, or simply return "I don't know" if the LLM decides that the corresponding chunk does not contain sufficient information. The final answer is generated by majority voting, with the "I don't know" responses excluded.

For our analysis, let us assume for simplicity that $m_i = m$ for all $i \in [k]$, and focus on understanding how the accuracy of the overall algorithm is impacted by the hyperparameter $m$, i.e. the chunk size. Following the approach in Section 2.3, this is achieve by first understanding each individual LLM call. We start by identifying two failure modes of each LLM call for retrieval from one chunk: (1) while the needle is contained in the chunk, the LLM might mistakenly return "I don't know" or an incorrect passcode; (2) while the chunk contains no needle, the LLM might hallucinate and mistakenly return a passcode that it believes is the true answer, especially when the chunk contains some objects that seem similar to the targeted object (e.g. "red lock" versus "red door"). It is reasonable to expect that the first failure mode will occur more frequently for a larger chunk size $m$; for the second failure mode, however, we observed empirically that some LLMs are more prone to it even when the value of $m$ is small, while others are much less so. Based on these observations, we hypothetically categorize LLMs into two types: *Type-1 LLMs* are only prone to the first failure mode, while *Type-2 LLMs* are prone to both. Now we are ready to consider the accuracy of the overall algorithm. With a Type-1 LLM, a smaller value of $m$ means the first failure mode is less likely to occur in Step 2 of the algorithm, which implies higher accuracy for the final solution. Analysis with a Type-2 LLM, on the other hand, is more complicated: a larger $m$ means the first failure mode is more likely to occur, while a smaller $m$ implies a larger number of chunks $k \asymp n/m$, which can potentially increase the chance of error in the final step of majority voting, due to the frequent occurrence of the second failure mode in Step 2 of the algorithm. Consequently, the minimum error of the overall algorithm might be obtained by some intermediate value of $m$ that achieves a balance between these two failure modes. If the input size $n$ is too large, then there might not exist a good value of $m$ that can achieve a low error, as either failure mode must occur with high probability.

## 4 HIERARCHICAL DECOMPOSITION

In this section, we apply the proposed analytical framework to the design and analysis of LLM-based algorithms following a more expressive pattern named *hierarchical decomposition*, where the original task is decomposed into multiple sub-tasks, and each of them can be further decomposed into more lower-level sub-tasks. We outline our study for this pattern and highlight the key insights in the following; the full version of this section can be found in Appendix D, which also includes empirical validation of our analysis.

**Task.** The concrete example under consideration is a question-answering task that requires retrieval of multiple needles from a large haystack and multi-hop reasoning over them. Suppose that the targeted question is about finding the numeric value of a particular variable, while the haystack consists of clues about the dependency between many variables, akin to the problem formulation considered by Ye et al. (2024). The needles embedded in the haystack are logically related, and some of the needles are related to the targeted question only *indirectly* via connection to other needles. For example, the targeted question might be "What is the numeric value of A?", while the needles are "A = B", "B = C", and "C = 100", located separately in different chunks of the haystack. An algorithm with the pattern of parallel decomposition, e.g. retrieving the needles from the chunks *independently* and then aggregating them for answering the targeted question, is doomed to fail in this case.

**Algorithm.** We consider an LLM-based algorithm that involves *multiple rounds of iterative retrieval and reasoning*, visualized in Figure 13 in the appendix. It starts by dividing the input haystack of size $n$ into $k$ chunks, each of size no larger than $m \asymp n/k$, and initializing an empty list of references for storing the relevant clues retrieved by LLM calls. Then for each round, the algorithm invokes $k$ sequential (Figure 13a) or parallel (Figure 13b) LLM calls for retrieval from $k$ chunks based on the targeted question and references, followed by one LLM call for reasoning about the updated references and deciding whether the algorithm is ready to answer the question or need more rounds of retrieval and reasoning. Similar approaches have been widely adopted in prior works (Creswell et al., 2023; Xiong et al., 2024; Qwen-Team, 2024). The resulting algorithm

exhibits a hierarchical structure: the original task is decomposed into multiple sequential rounds, and each round is further decomposed into multiple steps of retrieval and reasoning.

**Analysis and insights.** We are particularly interested in two critical design choices: the option of prompting the LLM calls responsible for reasoning and answering ("answer directly" or "think step by step"), and the option of retrieval from multiple chunks within each round (sequentially or in parallel). In the following, we let $r$ denote the number of rounds (which is determined adaptively by the algorithm itself at runtime), and $\ell$ denote the total length of the needles. We assume that, when the LLM is given all needles and asked to answer the targeted question, the number of generated tokens $\mathcal{L}_{\text{dec}}$ will be $O(1)$ if it is prompted to answer directly, or $O(\ell)$ if it is prompted to think step by step before answering. The cost of such an LLM call is thus bounded by $\mathcal{C}_{\text{LLM}}(\mathcal{L}_{\text{sys}} + \ell, 1 \text{ or } \ell)$.

For the case of sequential retrieval within each round (Figure 13a), we conclude that the total cost of the overall algorithm can be bounded as follows (see Appendix D.3 for the derivation):

$$\mathcal{C} \le r \times \Big( k \times \mathcal{C}_{\text{LLM}}(\mathcal{L}_{\text{sys}} + m + \ell, \ell) + \mathcal{C}_{\text{LLM}}(\mathcal{L}_{\text{sys}} + \ell, 1 \text{ or } \ell) \Big). \tag{5}$$

In particular, this bound quantifies how the cost of LLM calls for reasoning and answering, namely $r \times \mathcal{C}_{\text{LLM}}(\mathcal{L}_{\text{sys}} + \ell, 1 \text{ or } \ell)$, only occupies a small fraction of the total cost. These results also holds for the case of parallel retrieval (Figure 13b), but a better bound can be obtained for the end-to-end latency with parallelism degree $p$, by replacing the $k$ factor on the right-hand side of Eq. (5) with $\lceil k/p \rceil$. Note that the concrete value of $r$ in the case of parallel retrieval, where the list of references is updated only once at the end of each round, is typically larger than that in the case of sequential retrieval, where the list of references is updated immediately after each retrieval step.

We derive two major insights from the above analysis. (1) LLM nodes for reasoning and answering only occupy a small fraction of the total cost, while playing a critical role in the output (and thus accuracy) of the overall algorithm. Our general recommendation is thus prompting them to think step by step before answering, which will boost the accuracy significantly with minor loss in efficiency. (2) Each option of retrieval has its own pros and cons. The sequential option requires fewer rounds of retrieval and reasoning (thanks to the timely updates to the references), with the downside that all retrieval steps have to be executed sequentially. In contrast, the parallel option requires more rounds and thus larger costs, but can leverage parallelism for achieving a smaller end-to-end latency.

## 5 RECURSIVE DECOMPOSITION

This section studies *recursive decomposition*, a pattern that is vastly different from those in previous sections, yet still covered by our proposed framework. A recursive LLM-based algorithm starts from the original task and recursively generates intermediate sub-tasks, each of which can be solved by aggregating the solutions to its own children tasks. In particular, decomposing and/or solving each sub-task can be achieved by LLM calls, while the outline of recursive task decomposition remains symbolic. Such LLM-based algorithms have been widely applied in the literature (Kazemi et al., 2023; Schlag et al., 2023; Prasad et al., 2024; Schroeder et al., 2024; Lee & Kim, 2023; Khot et al., 2023). We outline our study for this pattern and highlight some key results in the following; the full version of this section (including empirical validation) can be found in Appendix E.

**Task.** We consider the same task from Section 4, which is about calculating a targeted variable based on clues about the dependency between many variables. One major difference here is that we consider a much larger number of relevant variables. The complex reasoning required to answer the targeted question correctly, even if all relevant clues are given *a priori*, can be well beyond the capability of one single LLM call, which motivates decomposing the reasoning process into multiple LLM calls. The other major difference is, we assume that the clues can be accessed only via querying a database: for each query, the database takes a name as input (say "A"), and returns a clue for the variable of the same name, e.g. "A = B + C" if A is a non-leaf variable, or "A = 10" if A is a leaf variable. Such a setting is motivated by (and can be regarded as an abstraction of) real-world scenarios where an autonomous agent in the wild need to actively retrieve relevant information by itself, via querying a real database, using a search engine, retrieving from documents, etc.

**Algorithm.** The key behind our algorithm design is a function named `ProcessNode`, which takes the name of a variable as input and returns its numeric value. This function is defined

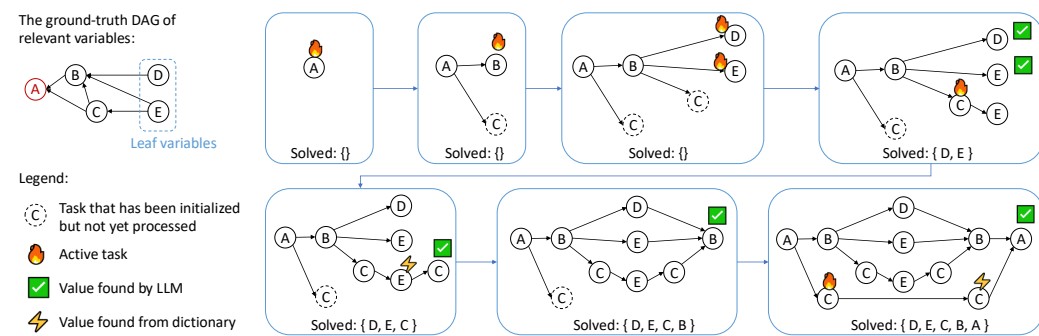

Figure 4: The computational graph of a recursive LLM-based algorithm is constructed dynamically, while a dictionary of solved sub-tasks is maintained to avoid solving the same sub-task repetitively.

recursively: if the value of the input variable can be easily found (e.g. it has been found before, or can be easily inferred from its corresponding clue returned by the database), then this value is returned; otherwise, one LLM call is invoked for spawning children tasks (i.e. identifying variables whose values will be useful for calculating the input variable), each of which is processed with one function call of `ProcessNode`, and finally the solutions to these children tasks are aggregated by an LLM call for solving the current task. It is worth noting that the computational graph of this algorithm, or more general LLM-based algorithms with recursive decomposition, is constructed dynamically in a depth-first-search style at runtime. See Figure 4 for a visualization.

**Analysis and a generic error bound.** We focus on analysis of accuracy here. For the current case study, errors of the algorithm largely arise from the limited arithmetic capabilities of LLMs. Moreover, all mathematical operations considered in this task, e.g. "A = B + C" or "A = max{B, C}", are assumed to be continuous with respect to each input variable. These motivate us to derive the following error bound, which indeed holds true for generic tasks and LLM-based algorithms satisfying the technical assumption explained in this proposition:

**Proposition 1.** *Suppose that the assumption of* additive errors and bounded sensitivity *holds for an LLM-based algorithm represented by a directed acyclic graph (DAG), i.e., for each node $v$ with inputs $x_1, \ldots, x_k$ and a single output $y$, it holds that $\mathcal{E}(y) \leq \mathcal{E}_v + S \times \sum_{i \in [k]} \mathcal{E}(x_i)$ for some node-specific additive error $\mathcal{E}_v$ and finite sensitivity parameter $S \geq 0$. Then the error of the output $y(v)$ of any node $v$, including the one that generates the output of the overall algorithm, is bounded by*

$$\mathcal{E}(y(v)) \leq \sum_{w \in DAG} \sum_{path \in \mathcal{P}(w \to v)} S^{|path|} \times \mathcal{E}_w, \tag{6}$$

*where $|path|$ denotes the length of a path on the DAG, and $\mathcal{P}(w \to v)$ represents the set of paths from node $w$ to node $v$ if $w \neq v$, or a set containing one hypothetical path of length 0 if $w = v$.*

## 6  CONCLUSION AND DISCUSSION

This work introduces an analytical framework for studying the design and analysis of LLM-based algorithms. After identifying the computational-graph representation, task decomposition as the design principle, and other abstractions, we find it feasible to provide formal analysis for the accuracy and efficiency of generic LLM-based algorithms. Through extensive case studies, we demonstrate the proposed framework in action and derive novel insights for various scenarios.

Due to limited space, we defer extended discussion on related works to Appendix A, and discussion on potential directions for future work to Appendix B. Appendix C includes additional examples for parallel decomposition studied in Section 3, while Appendices D and E are the full versions of Sections 4 and 5, which investigate hierarchical and recursive decomposition respectively.

Moving forward, we find it promising future research to further expand or apply the proposed framework and thereby advance the field of LLM-based algorithms, from a theoretical or practical perspective. We would like to invite the community to contribute to this exciting and rapidly developing field, using the current work as a starting point.

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

## A    RELATED WORKS

**LLM-based algorithms.**    The concept of LLM-based algorithms, as was explained in Section 1, is fairly broad and general, ranging from a combination of one or multiple LLM calls with some prompt engineering (Yao et al., 2023a; Besta et al., 2023; 2024; Lei et al., 2023; Saha et al., 2023; Zhou et al., 2023; Wang et al., 2023; Prasad et al., 2024; Chang et al., 2024; Zhang et al., 2023; 2024b), to LLM-powered agent systems (Mialon et al., 2023; Wu et al., 2023; Hong et al., 2024; Gao et al., 2024a; Shen et al., 2023; Zhuge et al., 2024; Pezeshkpour et al., 2024; Xi et al., 2023; Qian et al., 2024; Kapoor et al., 2024) and compound AI systems (Zaharia et al., 2024; Chen et al., 2024) that augment LLMs with additional abilities like tool use and long-term memory, and to the emerging paradigm of LLM programming (Schlag et al., 2023; Khot et al., 2023; Khattab et al., 2024; Zheng et al., 2024; Kambhampati et al., 2024).    Some elements of our analytical framework proposed in Section 2, such as the principle of task decomposition, the computational-graph representation, evaluation of and comparison between LLM-based algorithms with accuracy and efficiency taken into account simultaneously, etc., have already appeared in one way or another in these prior works. It is primarily our *unified, systematic and formal* investigation into the design and analysis of *generic* LLM-based algorithms that distinguishes the current work from this vast literature.

**Scaling properties of LLM test-time computation.**    Recent works have started to investigate, analytically or empirically, the scaling properties of LLM test-time computation.

One line of research is concerned about the scaling properties of repeated sampling, e.g. randomly generating multiple sequences for the same prompt and then aggregating the results (Chen et al., 2024; Brown et al., 2024; Snell et al., 2024; Wu et al., 2024), which might be regarded as a stochastic version of parallel decomposition investigated in Section 3.    Our work is orthogonal and complementary to this line of research, as our analysis is deterministic and targeted at generic patterns of LLM-based algorithms.    Indeed, one potential direction for expanding the analytical framework proposed in this work would be to augment it with stochastic decoding and repeated sampling, along with relevant theoretical results from prior works.

Another line of works is concerned about improving the output quality of an LLM call, albeit at a higher cost, by generating more tokens autoregressively, e.g. via chains of thoughts (Wei et al., 2022b) or step-by-step reasoning (Kojima et al., 2022). This idea has been investigated theoretically (Feng et al., 2023; Merrill & Sabharwal, 2024; Li et al., 2024b), and further popularized recently by the OpenAI o1 model (OpenAI, 2024a). As this approach of scaling up the test-time computation of *individual* LLM calls is becoming more widely adopted, it is important to understand, analytically and quantitatively, how it will impact the *overall* accuracy and efficiency when such LLM calls are embedded within an LLM-based algorithm. Our work can be useful in achieving this, as illustrated in the case studies in Appendices D and E, where the impacts of prompting LLM calls to "think step by step" versus "answer directly" can be characterized analytically with our proposed framework.

**Learning-augmented algorithms.**    Our work draws inspiration from the research area of algorithms with predictions / learning-augmented algorithms (Mitzenmacher & Vassilvitskii, 2022; Lindermayr & Megow, 2022).    One standard paradigm of this area is to consider a specific task (say sorting (Bai & Coester, 2023) or clustering (Ergun et al., 2022)), assume access to a black-box machine learning model that satisfies certain properties (e.g. what additional computation or information it can offer), propose a novel algorithm that leverages this ML model, provide theoretical guarantees for its accuracy and efficiency, and show the improvements over traditional, purely symbolic algorithms.    Our work is similar in spirit to this line of research, but also substantially different, in that our analytical framework is targeted at general tasks and LLM-based algorithms, with assumptions on the capabilities, limitations and inference service of LLMs (regarded as general-purpose problem solvers) that are quite different from typical assumptions for task-specific ML models in the literature of learning-augmented algorithms.

## B Further discussion

### B.1 Directions for future work

Moving forward, we find it promising future research to further expand or apply the proposed framework and thereby advance the field of LLM-based algorithms, from a theoretical or practical perspective. For example:

- We have been assuming that LLM nodes are state-less, and only considering the simplest and most straightforward usage of LLMs, i.e. sending a prompt and receiving a response for each LLM call. One potential extension is to consider LLM nodes with states, like short-term memory in agent systems, or other advanced ways of using LLMs, and understand how accuracy and efficiency of LLM-based algorithms are impacted in such cases.

- In our empirical study, we have mainly focused on the hyperparameter $m$ that indicates the granularity of parallel task decomposition, or the option of prompting certain LLM calls to either answer directly or think step by step. Future work might place more emphasis on other configurations of LLM-based algorithms, such as the choices of LLM models (e.g. choosing a strong-but-expensive model for specific LLM calls and a weak-but-cheap one for the others within the same algorithm), LLM decoding methods (e.g. greedy versus stochastic (Wang et al., 2023; Chen et al., 2024; Brown et al., 2024)), etc., and better understand how they impact the accuracy and efficiency of LLM-based algorithms.

- Through our abstractions of multiple error and cost metrics for the same task, we have touched upon the topics of multi-objective optimization and hyperparameter optimization. Future work might try to formally investigate these aspects.

- From a more practical perspective, future work might adopt the proposed framework, or some components and methodology within it, to assist the design, analysis, improvement and application of new LLM-based algorithms, or for fair comparison between different algorithms, with both accuracy and efficiency taken into account.

### B.2 Practical considerations

We make two comments about some practical considerations for the proposed framework.

First, regarding the capabilities and limitations of LLMs in a specific task, the black-box nature of LLMs can make it challenging to analytically and accurately quantify these factors, in which case one might resort to measuring and profiling in practice. On the positive side, it is oftentimes easier to make certain qualitative assumptions. For example, in many tasks of interest, a larger problem instance is harder than a smaller one, and thus incurs larger error metrics of an LLM call. For practical purposes like optimizing certain hyperparameters of an LLM-based algorithm, such weak assumptions might be sufficient already.

Second, understanding LLM inference service (Yuan et al., 2024; Pope et al., 2023; Zhou et al., 2024), especially from a system perspective, is crucial for in-depth analysis of cost metrics. LLM inference service can be diverse in practice: for example, LLMs might run on CPUs in a personal laptop or on a distributed GPU cluster, inference might be compute-bound or memory-bound, the complexity of long-sequence processing and generation might be linear or quadratic, parallelism at various levels (e.g. multiple LLMs deployed on multiple machines, or batch inference with one LLM) might be supported, and so on. All these are covered by our proposed framework in a unified manner.

Table 1: A list of notations for analysis of parallel decomposition.

| Notation | Definition |
|---|---|
| $n$ | Size of the input problem instance |
| $m$ | Size of each parallel sub-task, $m \leq \overline{m}$ |
| $k$ | Number of parallel sub-tasks, $k \lesssim n/m$ |
| $p$ | Maximum degree of parallelism |
| $\mathcal{L}_{\mathsf{sys}}$ | Maximum length of the system prompt |

## C  PARALLEL DECOMPOSITION (EXTENDED)

This section is dedicated to the analytical and empirical study of a few specific tasks and their corresponding LLM-based algorithms that follow the pattern of parallel decomposition. For convenient reference, Table 1 includes a list of notations introduced in Section 3.1.

**Concrete examples.**  Tasks that we consider in this section include counting, sorting, retrieval, retrieval-augmented generation (RAG) and long-text summarization, whose details will be explained in their corresponding subsections. For each task, we specify the concrete LLM-based algorithm, analyze its performance in terms of error and cost metrics, and validate our analysis with numerical experiments. Error metrics are task-specific, while cost metrics of interest are common among tasks, including the total prefilling length and decoding length, the total number of LLM calls, and the end-to-end latency with sequential or parallel LLM calls. These concrete examples not only confirm the practical advantages of LLM-based algorithms, but also verify that our analysis can help explain or predict the empirical performance of LLM-based algorithms, reveal the reasons behind some interesting empirical phenomena, and instruct the design of algorithms or choices of hyperparameters, e.g. the sub-task size $m$.

*Remark* 1. While the tasks under consideration are motivated by practical scenarios, our study will mostly focus on synthetic task design, like many prior works do. This brings numerous benefits, such as avoiding data contamination, allowing full transparency and control over task configurations, and making the current work as self-contained as possible.

**Experiment settings.**  We use the following LLMs in our experiments, which cover a wide range of LLM characteristics and inference service:

- A `Llama-3-8B` model (Meta, 2024), supported by ollama (ollama, 2023) and running on a Macbook Pro with a M2 Pro chip and 16GB memory;

- A `Llama-3-70B` model (Meta, 2024), supported by vLLM (Kwon et al., 2023) and running on a server with 4 Nvidia A100-80G GPUs;

- A `GPT-4-Turbo` model (OpenAI, 2024), accessed via API queries.

All of these LLMs are chat models. Each LLM call involved in our algorithms is prompted in a chat format, based on the sub-task that it is responsible for. Interested readers are referred to the source code for the prompts used in our experiments. We use greedy decoding, which is deterministic, in all experiments.

Below are a few more details about our experiments. (1) For ideal parallelism of LLM calls, we consider parallelism degree $p = 4$ and $p = \infty$. Latencies in the presence of parallelism are *simulated* according to Eq. (3). (2) In all experiments, the number of tokens for a piece of text is *estimated* using the same tokenizer, namely the `cl100k_base` encoding of the `tiktoken` package[3]. This simplification has no effect on the major conclusions from our experiment results. (3) Our experiment results include curves of some error metric (in blue) or cost metric (in red) versus the problem size $n$ or sub-task size $m$. For each curve, we plot the mean and standard deviation of measured metrics from multiple independent trials, i.e. multiple randomly generated task instances.

---

[3]`https://github.com/openai/openai-cookbook/blob/main/examples/How_to_count_tokens_with_tiktoken.ipynb`

## C.1 EXPERIMENTS FOR COUNTING

We validate our analysis in Section 3.2 for the counting example with numerical experiments. For each problem instance, the input string is generated by randomly sampling $n$ characters from the union of digits, lower-case letters and upper-case letters.

**Results with `Llama-3-8B`.** Our empirical results with a `Llama-3-8B` model are illustrated in Figure 5 and explained in the following.

In Figure 5a, we vary the problem size $n$ and set $m = n$ in our algorithm, which means the algorithm becomes equivalent to a single LLM call. Unsurprising, error metrics in this counting task monotonely increase with $n$. The number of prefilling tokens increases linearly with $n$, while the number of decoding tokens is insensitive to $n$, since we prompt the LLM to output its answer directly without intermediate reasoning steps. The latency of one LLM call also increases linearly with $n$, which is likely due to the relatively small sequence lengths and the compute-bound nature of LLM inference by a `Llama-3-8B` model running on a CPU.

In Figure 5b, we fix $n = 200$ and vary the hyperparameter $m$, namely the size of each sub-task in our proposed LLM-based algorithm. It is confirmed that decomposing the original task into smaller parallel sub-tasks with a smaller value of $m$ improves the accuracy of the overall algorithm, while incuring higher cost metrics, except for the latency with infinite parallelism (which is monotonely increasing in $m$) and the latency with parallelism degree $p = 4$ (which achieves the minimum at $m = n/p = 50$, as was predicted by our previous analysis).

**Results with `GPT-4-Turbo`.** Figure 6 demonstrates the empirical results for the same experiments but with a `GPT-4-Turbo` model. We observe from Figure 6a that the latency of one LLM call is insensitive to the input problem size $n$, which is likely because LLM inference of `GPT-4-Turbo` is memory-bound for the range of sequence lengths considered in our experiments. One potential implication is that, for the LLM-based algorithm with parallel decomposition, the latency with infinite parallelism $p = \infty$ might slightly increase for smaller $m$ and hence larger $k$, due to the random variation of latencies in reality; this is indeed what we observe from Figure 6b. Other than that, the results in Figure 6 are similar to those in Figure 5.

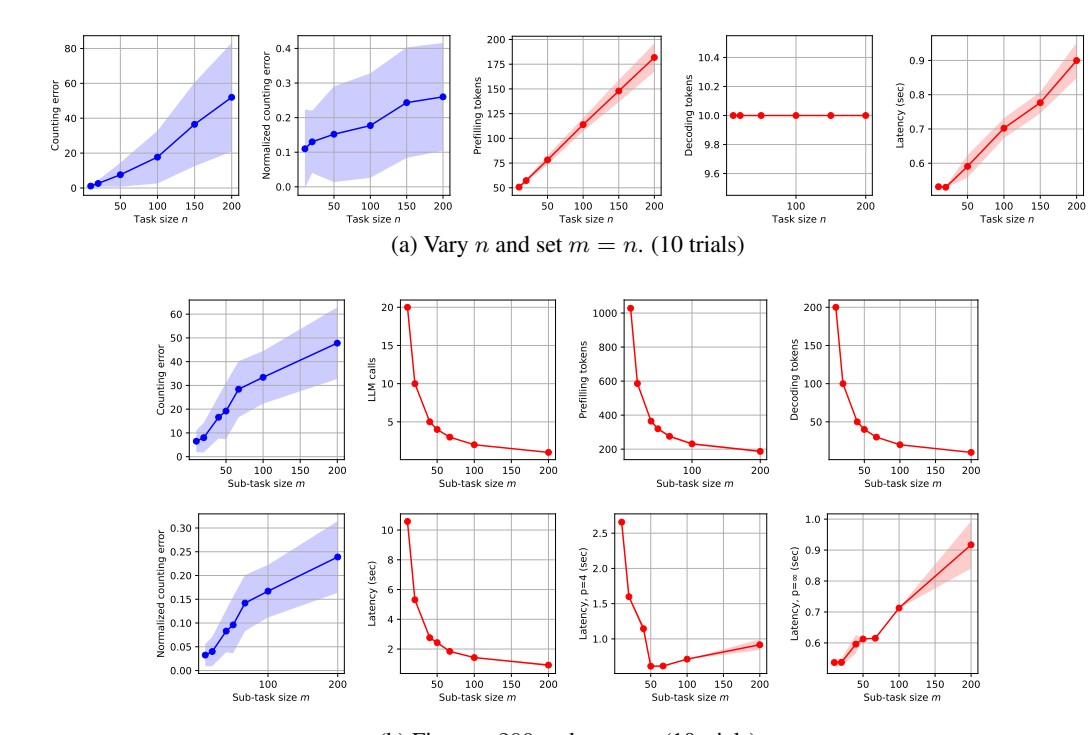

(a) Vary $n$ and set $m = n$. (10 trials)

(b) Fix $n = 200$ and vary $m$. (10 trials)

Figure 5: Empirical results for counting with `Llama-3-8B`.

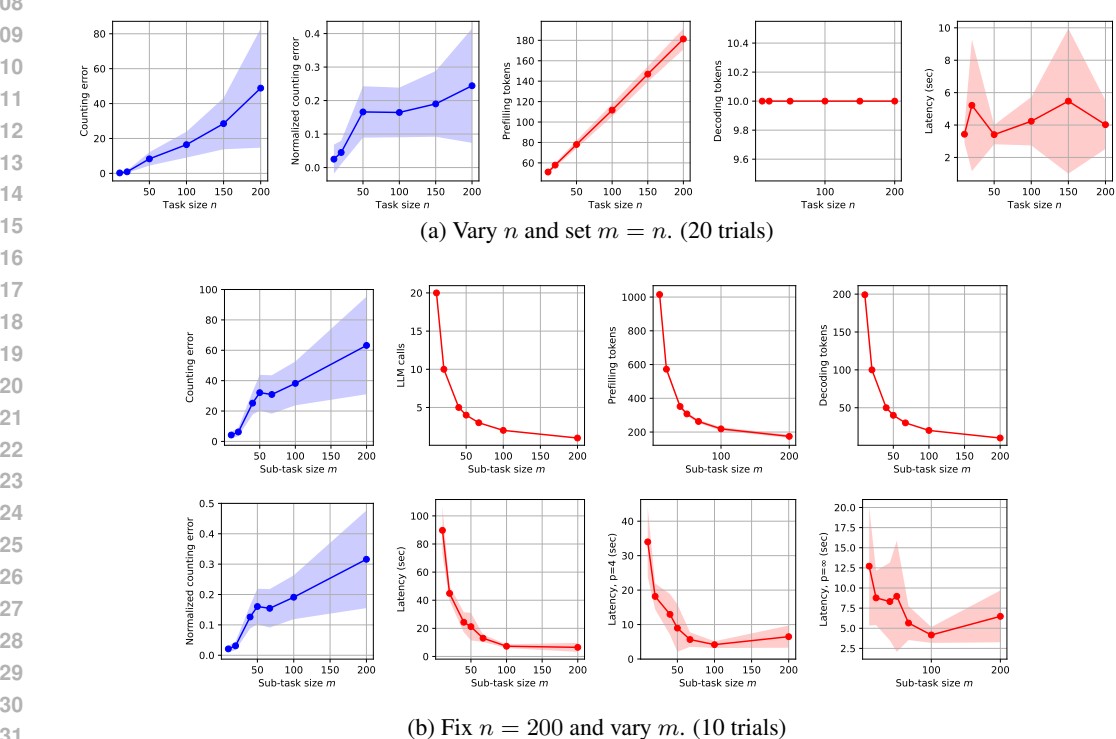

(a) Vary $n$ and set $m = n$. (20 trials)

(b) Fix $n = 200$ and vary $m$. (10 trials)

Figure 6: Empirical results for counting with `GPT-4-Turbo`.

## C.2 EXAMPLE: SORTING

For a more challenging example, let us consider the classical sorting problem: given a list of $n$ numbers $\boldsymbol{x} \in \mathbb{R}^n$, the task is to sort it in ascending order.

### C.2.1 ALGORITHM

Below is an LLM-based algorithm for sorting a list, which generalizes the naive approach of sorting the list with one single LLM call:

1. Divide the input list $\boldsymbol{x}$ into $k$ disjoint sub-lists $\boldsymbol{x}_1, \ldots, \boldsymbol{x}_k$ of lengths $m_1, \ldots, m_k$;

2. For each $i \in [k]$, use one LLM call to sort the $i$-th sub-list, which returns a solution $\boldsymbol{y}_i$;

3. Merge the sub-lists $\boldsymbol{y}_1, \ldots, \boldsymbol{y}_k$ into a single list $\boldsymbol{y}$ using a symbolic algorithm.

We note two details about this algorithm. (1) To ensure efficiency and stability, for each LLM call, we prompt the LLM to generate the sorted list directly, without intermediate reasoning steps. It has been verified empirically that the LLMs considered in our experiments can follow such instructions, and generate text of length $\mathcal{L}_{\mathsf{dec}}(m) = O(m)$ that can be easily parsed into a list. (2) Step 3 of the algorithm relies on a classical symbolic algorithm for merging two sorted lists, which maintains two moving pointers, one for each list, and chooses each entry of the merged list by comparing the values corresponding to the two pointers. Merging multiple sorted lists can be done by merging one pair of lists at a time, in an incremental or hierarchical manner. Python code for these procedures can be found in Listing 1 at the end of this subsection. Although they are designed under the assumption that the input lists are sorted, they can also be applied to input lists that are not fully sorted, which can possibly happen within the LLM-based algorithm since the input lists in Step 3 are generated by LLM calls in Step 2.

### C.2.2 ANALYSIS

Let us assume for notational convenience that $m_i = m$ for all $i \in [k]$, and $k = n/m$ is an integer.

**Error metrics.** Compared to counting, there are more diverse phenomena in the sorting task in terms of error metrics. In particular, multiple possible failure modes exist in sorting with an LLM-based algorithm:

1. The output list might not be monotone;

2. The length of the output list might be larger or smaller than that of the input list;

3. The output list might contain numbers that do not match exactly those of the input list.

Based on these failure modes, we define the following error metrics for sorting a list, where $\boldsymbol{y}$ denotes the solution returned by the algorithm and $\boldsymbol{y}^\star$ denotes the ground-truth solution:

- *Exact-match error*: $\mathcal{E} = 0$ if $\boldsymbol{y}$ matches $\boldsymbol{y}^\star$ exactly, and $\mathcal{E} = 1$ otherwise;
- *Non-monotonicity error*: $\mathcal{E} = \sum_{i \in [n-1]} \max\{y_i - y_{i+1}, 0\}$, which is zero if and only if $\boldsymbol{y}$ is perfectly sorted;
- *Length-mismatch error*: $\mathcal{E} = \frac{1}{n} |\mathsf{len}(\boldsymbol{y}) - \mathsf{len}(\boldsymbol{y}^\star)| = \frac{1}{n} |\mathsf{len}(\boldsymbol{y}) - n|$;
- *Fuzzy $\ell_\infty$ and fuzzy normalized $\ell_1$ errors*: we first convert, via simple extending or truncating, the output solution $\boldsymbol{y}$ to a version $\widehat{\boldsymbol{y}}$ that matches the length $n$ of the input list, and then calculate the fuzzy $\ell_\infty$ error as $\mathcal{E} = \|\widehat{\boldsymbol{y}} - \boldsymbol{y}^\star\|_\infty = \max_{i \in [n]} |\widehat{y}_i - y_i^\star|$, or the fuzzy normalized $\ell_1$ error as $\mathcal{E} = \frac{1}{n} \|\widehat{\boldsymbol{y}} - \boldsymbol{y}^\star\|_1 = \frac{1}{n} \sum_{i \in [n]} |\widehat{y}_i - y_i^\star|$.

Note that the same error metrics can be similarly defined for each parallel sub-task in Step 2 of the LLM-based algorithm, and it is reasonable to expect that they become smaller as the sub-task size $m$ decreases. On the other hand, analyzing the error metrics of the overall algorithm after the final merging step can be more complicated, and might be an interesting theoretical problem on its own. As an example, focusing on the third failure mode and the $\ell_\infty$ error metric, we have the following guarantee, whose proof is deferred after the experiments in this subsection.

**Proposition 2.** *Assume that for each $i \in [k]$, the solution $\boldsymbol{y}_i$ returned by one LLM call for the $i$-th sub-task is monotone, matches the length of the corresponding input $\boldsymbol{x}_i$, and has an $\ell_\infty$ error $\mathcal{E}_i$. Then the $\ell_\infty$ error of the final solution $\boldsymbol{y}$ is upper bounded by $\mathcal{E} \leq \max\{\mathcal{E}_1, \ldots, \mathcal{E}_k\}$.*

**Cost metrics.** Our analysis of cost metrics for the LLM-based sorting algorithm follows Section 3.1, and is similar to that for counting. One major difference is that $\mathcal{L}_{\mathsf{dec}} = O(m)$ rather than $O(1)$ for each sub-task.

- Considering the total prefilling length and decoding length as cost metrics, one has

$$\mathcal{C}(\text{prefilling}) \lesssim k \times (\mathcal{L}_{\mathsf{sys}} + m) = \frac{n}{m} \times (\mathcal{L}_{\mathsf{sys}} + m) = n \times \left(\frac{\mathcal{L}_{\mathsf{sys}}}{m} + 1\right),$$

$$\mathcal{C}(\text{decoding}) \lesssim k \times m = n.$$

  The former is decreasing in $m$, while the latter is insensitive to $m$.

- More generally, the sum of costs of all LLM calls is

$$\mathcal{C} = \mathcal{C}(\text{sub-tasks}) \leq k \times \Big(\mathcal{C}_{\mathsf{pre}}(\mathcal{L}_{\mathsf{sys}} + m) + \mathcal{C}_{\mathsf{dec}}(\mathcal{L}_{\mathsf{sys}} + m, m)\Big)$$

$$= n \times \frac{\mathcal{C}_{\mathsf{pre}}(\mathcal{L}_{\mathsf{sys}} + m) + \mathcal{C}_{\mathsf{dec}}(\mathcal{L}_{\mathsf{sys}} + m, m)}{m}.$$

- For the end-to-end latency with parallelism degree $p$, we have

$$\mathcal{C} = \mathcal{C}(\text{sub-tasks}) \leq \left\lceil \frac{k}{p} \right\rceil \times \Big(\mathcal{C}_{\mathsf{pre}}(\mathcal{L}_{\mathsf{sys}} + m) + \mathcal{C}_{\mathsf{dec}}(\mathcal{L}_{\mathsf{sys}} + m, m)\Big)$$

$$= \left\lceil \frac{n}{p \times m} \right\rceil \times \Big(\mathcal{C}_{\mathsf{pre}}(\mathcal{L}_{\mathsf{sys}} + m) + \mathcal{C}_{\mathsf{dec}}(\mathcal{L}_{\mathsf{sys}} + m, m)\Big).$$

C.2.3 EXPERIMENTS

We validate our analysis with numerical experiments. The input list of each problem instance is generated by randomly sampling entries of the list from the uniform distribution over the interval $[0, 1]$ and then rounding each of them to two decimals.

**Results with `Llama-3-70B`.** Our empirical results with a `Llama-3-70B` model are illustrated in Figure 7 and explained in the following.

In Figure 7a, we vary the problem size $n$ and set $m = n$ in the LLM-based algorithm, in which case the algorithm becomes equivalent to a single LLM call. We make the following observations:

- Unsurprisingly, all error metrics in this task monotonely increase with $n$.
- While the LLM might output a list that deviates from the ground-truth solution, it is at least good at ensuring that the output list itself is sorted or has a very small non-monotonicity error.
- The prefilling length grows linearly with $n$, while the growth of the decoding length and end-to-end latency slows down slightly for large values of $n$, which is mainly because the LLM is prone to returning a list that is shorter than the input list when $n$ is large, as reflected in the length-mismatch error curve.

In Figure 7b, we fix $n = 200$ and vary the sub-task size $m$. It is confirmed that decomposing the original task into smaller parallel sub-tasks with a smaller value of $m$ implies lower error metrics achieved by the overall algorithm, while increasing certain cost metrics. Two specific observations:

- The total number of decoding tokens decreases with $m$ at a rate that matches the length-mismatch error curve. This does not contradict our previous analysis, which predicts an *upper bound* that is insensitive to the value of $m$.
- Regarding the end-to-end latency with parallelism degree $p = 4$, the zigzag part of the curve might seems curious. In fact, a fine-grained analysis can well explain this phenomenon. If we approximate the latency for one LLM call solving a sub-task of size $m$ by $O(m)$, then the end-to-end latency of the overall algorithm is approximately $O(m \times \lceil k/p \rceil)$. The numbers calculated in Table 2 for the concrete setting of this experiment match the empirical results and explain the zigzag part.

Table 2: Fine-grained analysis for the latency with parallelism degree $p = 4$ in the setting of Figure 7b, where $n = 200$.

| Sub-task size $m$ | 10 | 20 | 40 | 50 | 67 | 100 | 200 |
|---|---|---|---|---|---|---|---|
| Number of sub-tasks $k = \lceil n/m \rceil$ | 20 | 10 | 5 | 4 | 3 | 2 | 1 |
| Sequential depth $d = \lceil k/p \rceil$ | 5 | 3 | 2 | 1 | 1 | 1 | 1 |
| Predicted latency $\asymp d \times m$ | 50 | 60 | 80 | 50 | 67 | 100 | 200 |

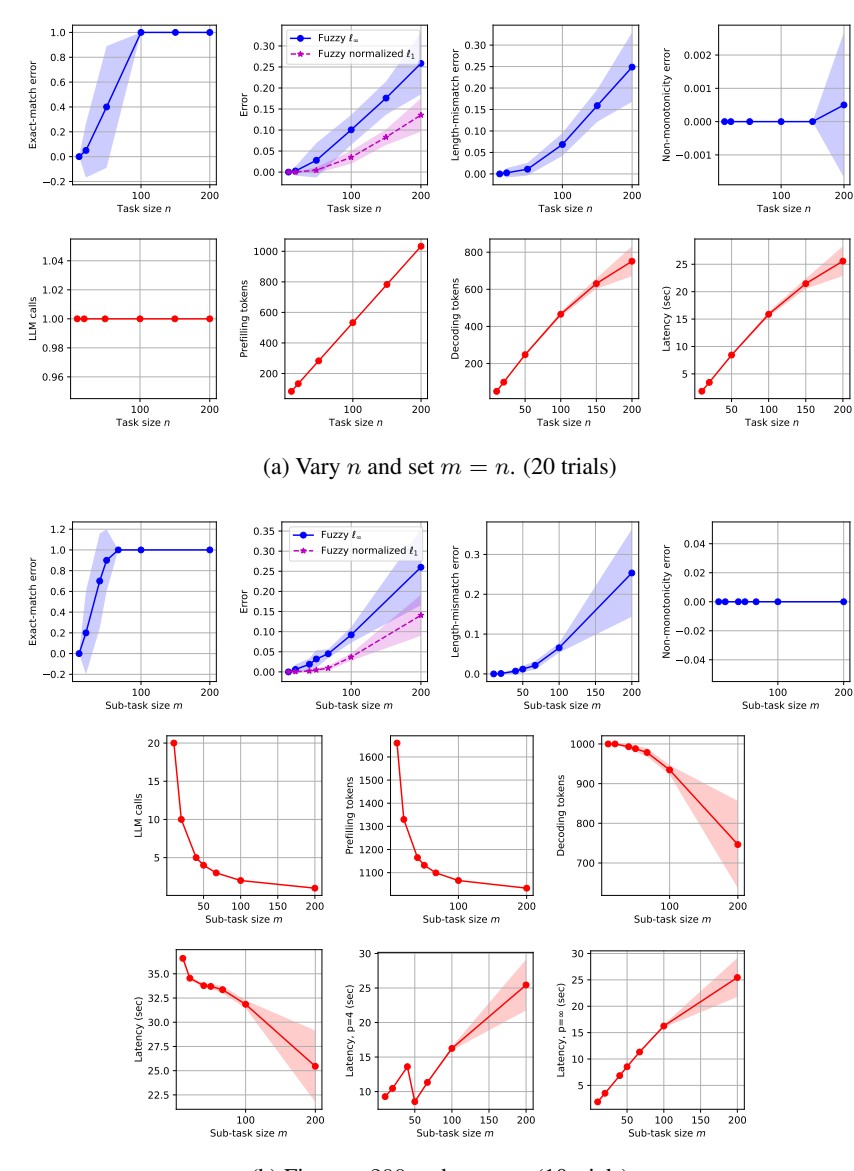

(a) Vary $n$ and set $m = n$. (20 trials)

(b) Fix $n = 200$ and vary $m$. (10 trials)

Figure 7: Empirical results for sorting with `Llama-3-70B`.

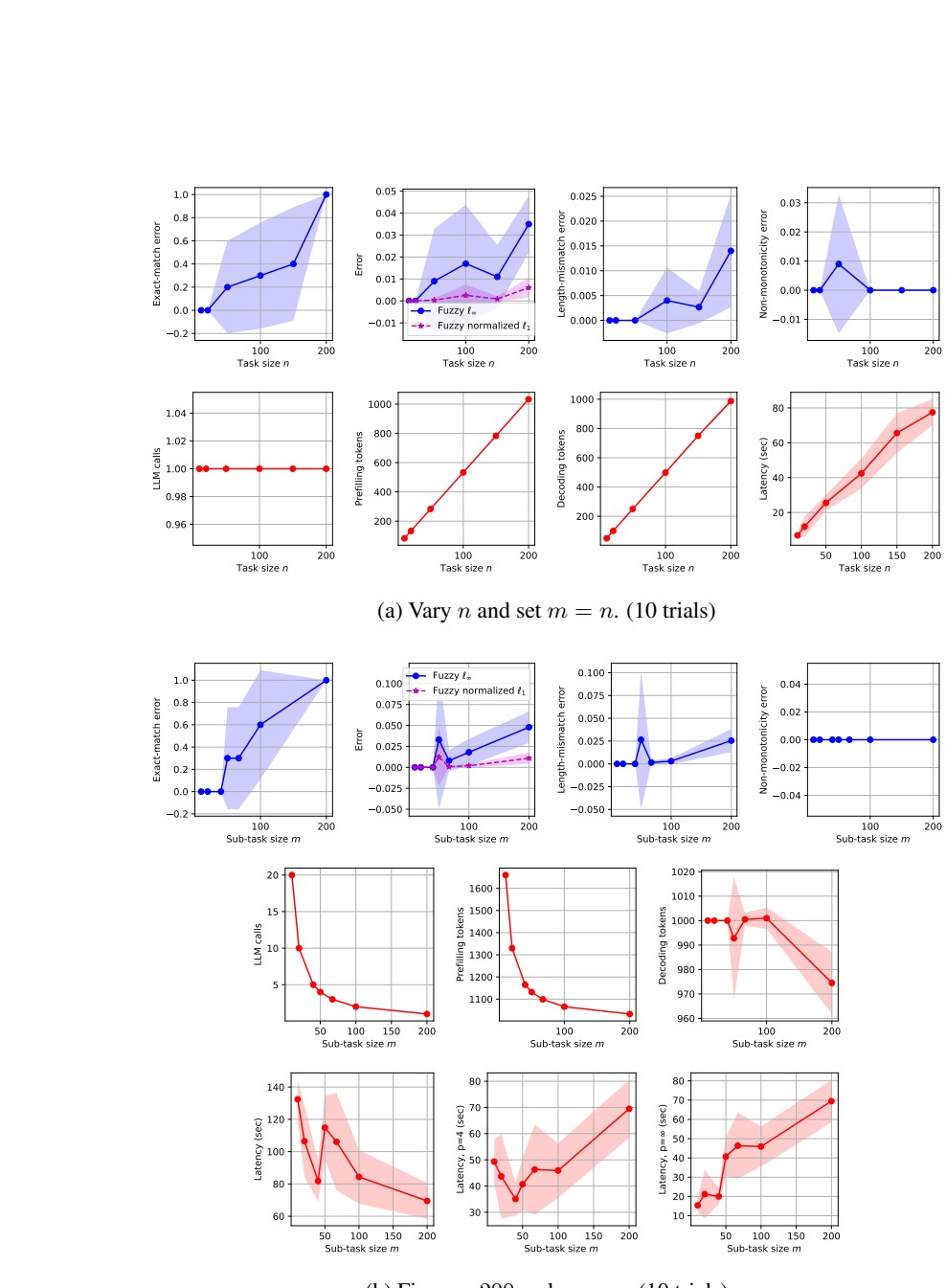

(a) Vary $n$ and set $m = n$. (10 trials)

(b) Fix $n = 200$ and vary $m$. (10 trials)

Figure 8: Empirical results for sorting with GPT-4-Turbo.

**Results with `GPT-4-Turbo`.** Figure 8 demonstrates the empirical results for the same experiments but with a `GPT-4-Turbo` model. Similar observations can be made from these results, except that latencies exhibit higher variance due to the inference service of `GPT-4-Turbo`.

### C.2.4 PROOF OF PROPOSITION 2

Recall that $\boldsymbol{y}_i$ denotes the solution returned by an LLM call, which is assumed to be monotone, for sorting the $i$-th input sub-list. Let $\boldsymbol{y}_i^\star$ denote the ground-truth solution for sorting the $i$-th input sub-list. Assuming that $\|\boldsymbol{y}_i - \boldsymbol{y}_i^\star\|_\infty \leq \epsilon$ for all $i \in [k]$, our goal is to prove that $\|\boldsymbol{y} - \boldsymbol{y}^\star\|_\infty \leq \epsilon$.

It is easy to check that merging multiple sorted lists is equivalent to sorting the concatenation of the lists. Therefore, we have

$$\boldsymbol{y} = \mathsf{sort}([\boldsymbol{y}_1, \ldots, \boldsymbol{y}_k]) = \mathsf{sort}(\mathsf{permute}([\boldsymbol{y}_1, \ldots, \boldsymbol{y}_k])),$$
$$\boldsymbol{y}^\star = \mathsf{sort}([\boldsymbol{y}_1^\star, \ldots, \boldsymbol{y}_k^\star]) = \mathsf{sort}(\mathsf{permute}([\boldsymbol{y}_1^\star, \ldots, \boldsymbol{y}_k^\star])),$$

where $\mathsf{permute}$ can be any permutation of $n$ elements in a list. In particular, we let $\mathsf{permute}$ be the permutation that sorts $[\boldsymbol{y}_1^\star, \ldots, \boldsymbol{y}_k^\star]$, which implies

$$\boldsymbol{y}^\star = \mathsf{sort}(\mathsf{permute}([\boldsymbol{y}_1^\star, \ldots, \boldsymbol{y}_k^\star])) = \mathsf{permute}([\boldsymbol{y}_1^\star, \ldots, \boldsymbol{y}_k^\star]).$$

Also notice that

$$\left\|\mathsf{permute}([\boldsymbol{y}_1, \ldots, \boldsymbol{y}_k]) - \mathsf{permute}([\boldsymbol{y}_1^\star, \ldots, \boldsymbol{y}_k^\star])\right\|_\infty = \left\|[\boldsymbol{y}_1, \ldots, \boldsymbol{y}_k] - [\boldsymbol{y}_1^\star, \ldots, \boldsymbol{y}_k^\star]\right\|_\infty$$
$$= \max_{i \in [k]} \|\boldsymbol{y}_i - \boldsymbol{y}_i^\star\|_\infty \leq \epsilon.$$

Based on the above analysis, our initial goal boils down to the following problem: given two lists $\boldsymbol{z}, \boldsymbol{z}^\star \in \mathbb{R}^n$ such that $\|\boldsymbol{z} - \boldsymbol{z}^\star\|_\infty \leq \epsilon$ and $\boldsymbol{z}^\star$ is sorted, we need to show that $\|\mathsf{sort}(\boldsymbol{z}) - \boldsymbol{z}^\star\|_\infty \leq \epsilon$. Here, $\boldsymbol{z}$ corresponds to $\mathsf{permute}([\boldsymbol{y}_1, \ldots, \boldsymbol{y}_k])$, and $\boldsymbol{z}^\star$ corresponds to $\boldsymbol{y}^\star$.

To prove this, let us consider the classical in-place insertion-sort algorithm illustrated in Algorithm 1. We choose to prove by induction that, throughout the execution of this algorithm where $\boldsymbol{z}$ is updated in place, it always holds that $\|\boldsymbol{z} - \boldsymbol{z}^\star\|_\infty \leq \epsilon$, which immediately implies $\|\mathsf{sort}(\boldsymbol{z}) - \boldsymbol{z}^\star\|_\infty \leq \epsilon$ for the initial $\boldsymbol{z}$. To prove this, notice that the only place in Algorithm 1 where $\boldsymbol{z}$ is changed is the step of swapping $z_j$ and $z_{j-1}$ when the condition $z_j < z_{j-1}$ is satisfied. Under this condition, we have the following for $\boldsymbol{z}$ before the swapping happens:

$$z_{j-1} - z_j^\star > z_j - z_j^\star \geq -\epsilon,$$
$$z_{j-1} - z_j^\star \leq z_{j-1} - z_{j-1}^\star \leq \epsilon,$$
$$z_j - z_{j-1}^\star < z_{j-1} - z_{j-1}^\star \leq \epsilon,$$
$$z_j - z_{j-1}^\star \geq z_j - z_j^\star \geq -\epsilon,$$

which implies $|z_{j-1} - z_j^\star| \leq \epsilon$ and $|z_j - z_{j-1}^\star| \leq \epsilon$. This means that the $\ell_\infty$ error bound is preserved after this swapping step, which concludes our proof.

---

**Algorithm 1:** The classical insertion-sort algorithm

1 **Input:** a list $\boldsymbol{z} \in \mathbb{R}^n$ to be sorted.
2 **for** $i = 2, 3, \ldots, n$ **do**
3     **for** $j = i, i-1, \ldots, 2$ **do**
4         **if** $z_j \geq z_{j-1}$ **then**
5             Break.
6         Swap $z_j$ and $z_{j-1}$.
7 **Output:** the sorted list $\boldsymbol{z}$.

---

```python
import numpy as np

def merge_two_sorted_lists(list1, list2):
    """Merge two non-empty lists or np.arrays that are
    assumed to be (at least approximately) sorted"""

    len1, len2 = len(list1), len(list2)
    idx1, idx2 = 0, 0
    idx = 0
    solution = np.zeros(len1 + len2)
    while idx < len1 + len2:
        if idx1 == len1:
            val = list2[idx2]
            idx2 += 1
        elif idx2 == len2:
            val = list1[idx1]
            idx1 += 1
        else:
            val1, val2 = list1[idx1], list2[idx2]
            if val1 <= val2:
                val = val1
                idx1 += 1
            else:
                val = val2
                idx2 += 1
        solution[idx] = val
        idx += 1

    return solution

def merge_sorted_lists_incremental(lists):
    """Merge lists = [list1, list2, ...] in an incremental manner"""

    for _ in range(len(lists) - 1):
        list1 = lists.pop()
        list2 = lists.pop()
        solution = merge_two_sorted_lists(list1, list2)
        lists.append(solution)

    return lists[0]

def merge_sorted_lists_hierarchical(lists):
    """Merge lists = [list1, list2, ...] in a hierarchical manner"""

    while len(lists) > 1:
        niters = len(lists) // 2
        for _ in range(niters):
            list1 = lists.pop(0)
            list2 = lists.pop(0)
            solution = merge_two_sorted_lists(list1, list2)
            lists.append(solution)

    return lists[0]
```

Listing 1: Python code for merging sorted lists. We choose the hierarchical option in our experiments.

### C.3 EXAMPLE: RETRIEVAL

We study a more realistic application of LLM-based algorithms, which is answering a given question that requires retrieving some key information from a long piece of text. Our design of this task draws inspiration from the needle-in-a-haystack benchmark (Kamradt, 2023) and other similar benchmarks that have been widely adopted for evaluating the long-context capability of LLMs as well as techniques of retrieval-augmented generation (Weston et al., 2015; Roucher, 2023; Kuratov et al., 2024; Mohtashami & Jaggi, 2023; Zhang et al., 2024a).

Consider the following setting as an example. A key message (the needle) of the form "The passcode to the {targeted object, e.g. red door} is {6-digit passcode}" is randomly inserted into a piece of long text (the haystack). The algorithm is asked to answer the question "What is the passcode to the {targeted object}?". To make the problem more challenging and fun, we let the haystack consist of alike sentences of the form "The passcode to the {colored object, e.g. red lock or green door} is {6-digit passcode}", with colored objects different from the targeted object. This allows us to investigate both sides of retrieval capabilities of LLMs and LLM-based algorithms: retrieving the targeted information correctly, while avoiding being confused or misled by background information that might seem relevant to the question (Shi et al., 2023).

Note that while we use this concrete setting for our empirical study, the proposed algorithm and analysis in the following are actually applicable to generic settings of this retrieval task.

#### C.3.1 ALGORITHM

We consider the following LLM-based algorithm that follows the pattern of parallel decomposition:

1. Divide the input text of length $n$ into $k$ overlapping chunks of lengths $m_1, \ldots, m_k$;

2. For each chunk, use one LLM call to try to answer the question based on that chunk;

3. Generate the final answer by majority voting.

We note a few details about this algorithm. (1) All lengths involved here are measured by the number of characters. (2) In the first step of chunking, we let each pair of adjacent chunks share an overlap that is larger than the length of the needle, to ensure that the needle will appears as a whole in at least one chunk. (3) For each LLM call in the second step, the LLM is prompted to answer "I don't know" if it believes that there is not sufficient information in the corresponding chunk, e.g. when the chunk simply does not contain the needle. Such answers will be excluded from the final step of the algorithm. (4) In the final step of majority voting, it is possible that there are multiple (say $h$) candidate solutions with the same frequency, in which case we let the algorithm return the list of such candidates. If this list contains the ground-truth solution, we calculate the exact-match error as $1 - 1/h$ in our experiments.

#### C.3.2 ANALYSIS

Let us assume for concreteness that each pair of adjacent chunks share an overlap of length $m/2$, and $m_i = m$ for all $i \in [k-1]$, while $m/2 \le m_k \le m$. In this case, we have $k = \lceil 2n/m - 1 \rceil$.

**Error metrics.** Let us focus on how error metrics of the LLM-based algorithm are impacted by the hyperparameter $m$. We start by identifying two failure modes of each LLM call for retrieving the targeted information from a chunk of size $m$ in Step 2 of the algorithm:

1. *The first failure mode* is that, while the needle is contained in the chunk, the LLM might mistakenly return "I don't know" or an incorrect passcode. It is reasonable to expect that this failure mode will occur more frequently for larger values of $m$. Our early experiments with various LLMs confirmed that this failure mode starts to occur when $m$ exceeds a certain threshold specific to each LLM.

2. *The second failure mode* is that, while the needle is actually absent from the chunk, the LLM might mistakenly return a passcode that it believes is the true answer to the question. We observed empirically that this is more likely to happen when the chunk contains some objects that seem similar to the targeted object (e.g. "red lock" or "green door" versus "red

door"), and that some LLMs are more prone to this failure mode even when the value of $m$ is small, while others are less so.

Based on the above observations, we can hypothetically categorize LLMs into two types: *Type-1 LLMs* are only prone to the first failure mode, while *Type-2 LLMs* are prone to both. It turns out that analysis of error metrics for the overall LLM-based algorithm is dependent on the type of the LLM being used.

- Analysis is simpler if a Type-1 LLM is used: a smaller value of $m$ means the first failure mode is less likely to occur in Step 2 of the algorithm, which implies higher accuracy for the final solution of the overall algorithm.

- Analysis is more complicated if a Type-2 LLM is used, since both failure modes can possibly occur in Step 2 of the algorithm. A larger value of $m$ means the first failure mode is more likely to occur, while a smaller value of $m$ implies a larger number of chunks $k = \lceil 2n/m - 1 \rceil$, which can potentially increase the chance of error in the final step of majority voting, due to the frequent occurrence of the second failure mode in Step 2 of the algorithm. Consequently, the minimum error of the overall algorithm might be achieved by some intermediate value of $m$ that achieves a balance between the two failure modes. If $n$ is too large, then there might not exist a good value of $m$ that can achieve a low error, as either failure mode must occur with high probability.[4]

**Cost metrics.** Our analysis of cost metrics for this task is largely the same as that for the counting task, despite how different these two tasks might seem. Under some mild conditions explained in Section 3.1, most cost metrics of interest are monotonely decreasing in $m$, except for the end-to-end latency with parallel LLM calls, which is increasing in $m$ for parallelism degree $p = \infty$ and possibly non-monotone for finite $p$. One thing to note is that, due to the overlaps between consecutive chunks, the number of parallel sub-tasks in Step 2 of the algorithm is $k = \lceil 2n/m - 1 \rceil$, rather than $\lceil n/m \rceil$ as in the counting task. This implies that the value of $m$ minimizing the latency can be predicted by letting $2n/m - 1 \approx p$, namely $m \approx 2n/(p + 1)$.

C.3.3  EXPERIMENTS

We validate our analysis with numerical experiments. For each task instance, the passcodes of all objects, the targeted object, the position of the haystack where the needle is inserted, etc., are all randomly chosen. The error metric of interest, namely the exact-match error, takes value $0$ if the final solution of the algorithm is exactly the same as the ground-truth passcode to the targeted object, $1 - 1/h$ if the algorithm returns a list of $h$ candidate solutions that includes the ground-truth passcode, and 1 otherwise.

**Results with `Llama-3-8B`.** Figure 9 includes the results of our experiments with `Llama-3-8B`.[5]

---

[4]An informal probabilistic analysis is as follows. Given the sub-task size $m$, denote the probability of the first and second failure modes as $p_1(m)$ and $p_2(m)$ respectively. Then, the success rate of retrieval for the chunk containing the needle is $1 - p_1(m)$, while the expected number of "false positives" from the remaining chunks is approximately $k \times p_2(m) \approx 2n \times p_2(m)/m$. One might opt for a relatively small value of $m$, which hopefully increases $1 - p_1(m)$ and hence mitigates the first failure mode. However, even if $p_2(m)/m$ is very small, say $10^{-3}$, the number of false positives can still be large if the size $n$ of the original problem is large, which will cause errors in the solution returned by majority voting.

[5]After executing many LLM calls in a row during our experiments with `Llama-3-8B` supported by ollama (ollama, 2023), we started to observe unusually large latencies (at least two orders of magnitude larger than their normal values) for some LLM calls, even though the generated texts are normal. We believe that this is most likely due to memory-related issues caused by running ollama on a laptop with limited 16GB memory, which can be easily avoided if a laptop with more memory is used. To mitigate this issue in our experiments, we take a different approach, i.e. adding a 3-second pause between each pair of consecutive LLM calls in the LLM-based algorithm when `Llama-3-8B` and ollama are used. While this proves to be quite effective, anomalies might still occur after running the experiments for a long period of time, in which case we simply re-run the part of experiments containing such anomalies, or remove these data points manually before plotting if there are very few of them.

In Figure 9a, we vary the length $n$ of the input text containing one needle in a haystack, and set $m = n$ for the LLM-based algorithm, which becomes equivalent to a single LLM call. Notice that the exact-match error, which corresponds to the first failure mode explained earlier, approaches zero as the problem size $n$ decreases. In addition, the decoding length is $O(1)$, and the wall-clock latency is $O(\mathcal{L}_{\mathsf{sys}} + n)$; one detailed observation is that as $n$ increases, it becomes more likely that the LLM call returns "I don't know", which slightly decreases the number of decoding tokens and thus the latency.

Figure 9b includes the results for the same experiment setting, except that no needle is inserted into the haystack. One crucial observation is that the exact-match error in this case, which corresponds to the second failure mode of retrieval, remains non-zero even for very small values of $n$. This suggests that `Llama-3-8B` should be regarded as a Type-2 LLM prone to both failure modes.

In Figures 9c and 9d, we vary the sub-task size $m$ while $n$ is fixed at 10000 and 20000 respectively. As was predicted by our analysis, the error of the overall algorithm is not monotone in the value of $m$, due to the presence of two failure modes. Another difference from the previous counting or sorting task is that the latency with parallelism degree $p = 4$ achieves the minimum around $m = 2n/(p+1) = 0.4n$ rather than $m = n/p = 0.25n$, which again matches our previous analysis.

**Results with `Llama-3-70B`.**  Figure 10 includes the results for the same experiments but with `Llama-3-70B` used within the LLM-based algorithm.

In particular, Figure 10a shows that `Llama-3-70B` achieves lower errors in the first failure mode of retrieval compared to `Llama-3-8B`, while Figure 10b suggests that `Llama-3-70B` is much less prone to the second failure mode and hence might be regarded as a Type-1 LLM. Consequently, in Figures 10c and 10d, the exact-match error exhibits a more monotone relation with the sub-task size $m$.

Regarding the cost metrics, results with `Llama-3-70B` are similar to those with `Llama-3-8B`.

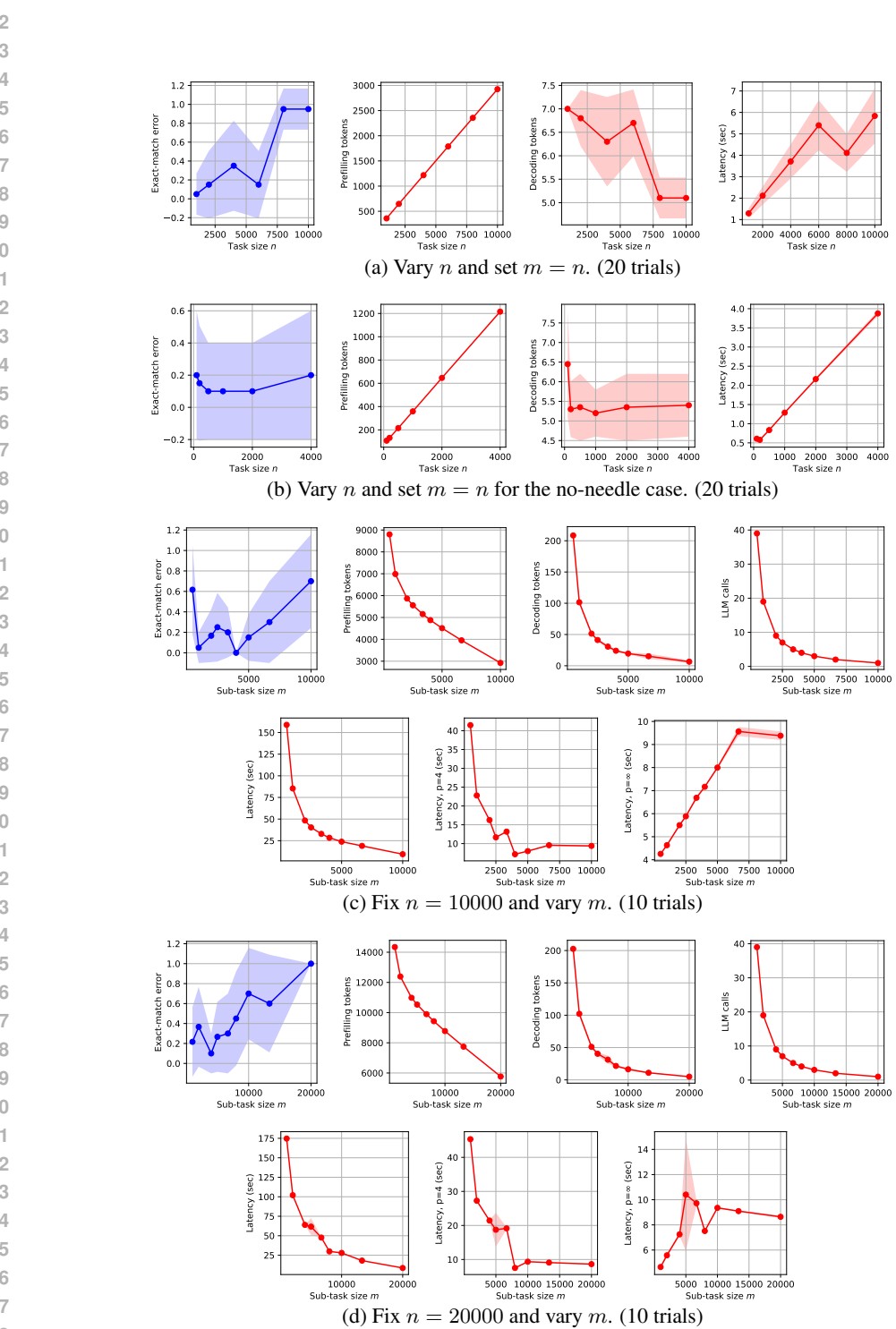

(a) Vary $n$ and set $m = n$. (20 trials)

(b) Vary $n$ and set $m = n$ for the no-needle case. (20 trials)

(c) Fix $n = 10000$ and vary $m$. (10 trials)

(d) Fix $n = 20000$ and vary $m$. (10 trials)

Figure 9: Empirical results for retrieval with `Llama-3-8B`.

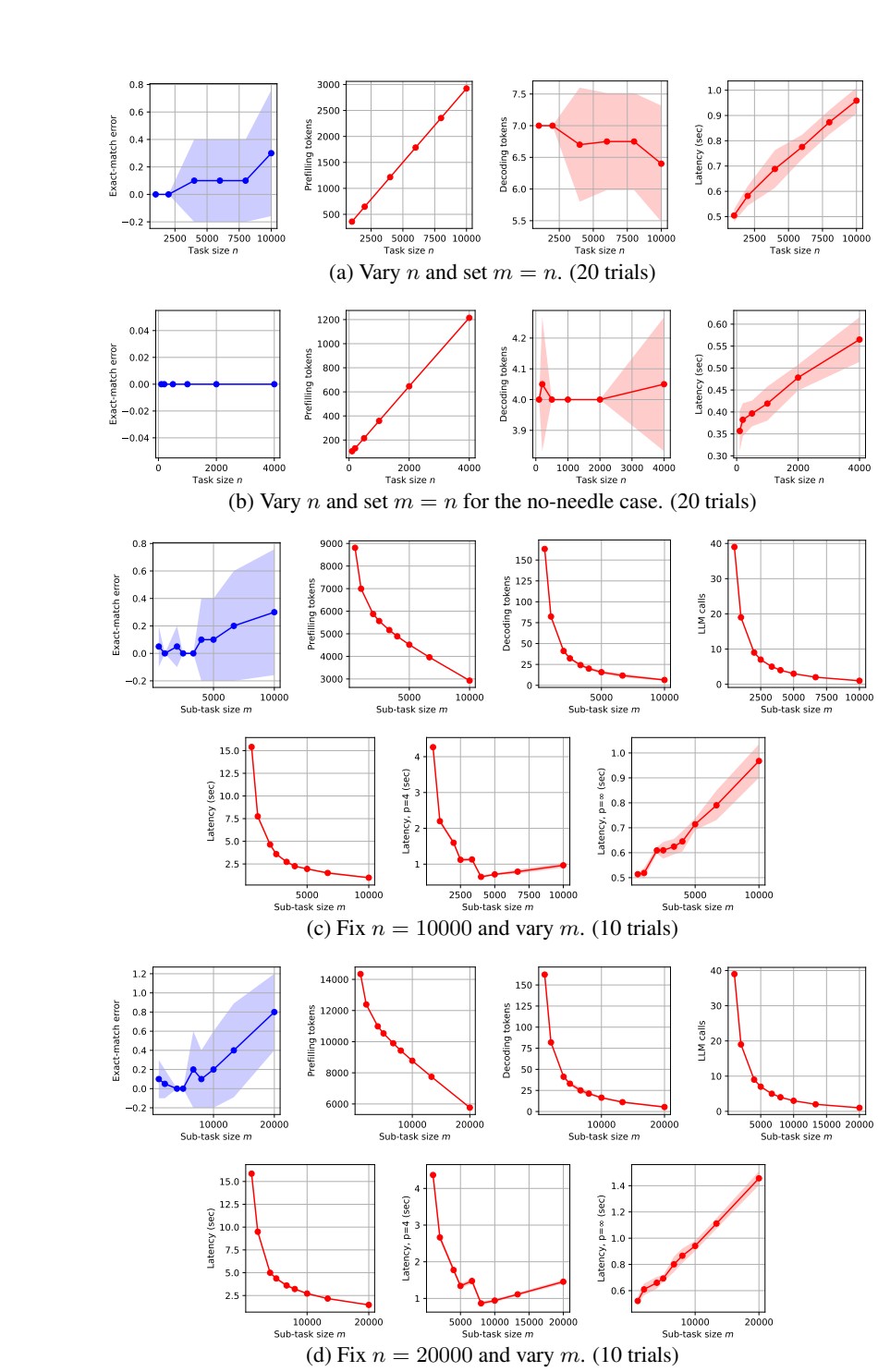

Figure 10: Empirical results for retrieval with `Llama-3-70B`.

## C.4 EXAMPLE: RETRIEVAL-AUGMENTED GENERATION

Our next example is a multiple-needle generalization of the previous retrieval task, which can be regarded as a synthetic and simplified version of retrieval-augmented generation (RAG) (Lewis et al., 2020; Gao et al., 2024b). In particular, we consider fine-grained sentence-level retrieval by LLMs, rather than document-level or chunk-level retrieval by certain similarity measure of dense embedding vectors.

More concretely, suppose that the input text is composed of sentences of the form "The {i}-th digit of the passcode to the {colored object}" is {digit}", where $i \in [6]$. The algorithm is asked to answer the question "What is the 6-digit passcode to the {targeted object}?". Compared with the previous single-needle retrieval task, here the algorithm need to retrieve multiple needles, each for one digit of the targeted object, in order to answer the question correctly; moreover, the final aggregation step requires certain capability of reasoning or summarization over the retrieved needles.

Note again that while we focus on this specific setting in our experiments, the algorithm and analysis in the following are actually applicable to generic settings of this RAG task.

### C.4.1 ALGORITHM

We consider the following LLM-based algorithm for solving this task:

1. Divide the input text of length $n$ into $k$ overlapping chunks of lengths $m_1, \ldots, m_k$;

2. For each chunk, use one LLM call to retrieve sentences that can be useful for answering the question;

3. Put the retrieved sentences together, based on which one LLM call is invoked for answering the question.

We note a few details about this algorithm. (1) For each chunk in Step 2, we prompt the LLM to retrieve relevant sentences, or return "None" if no relevant sentence is found in that chunk. Such "None" results will be excluded from Step 3 of the algorithm. (2) Unlike previous examples, the final aggregation step of this algorithm involves an LLM node, which adds to the cost metrics of the overall algorithm. (3) For simplicity, we assume that the number of needles (i.e. length of the passcode) and the length of each needle are both $O(1)$. For more general cases, the final aggregation step might benefit from further task decomposition. (4) We allow the algorithm to return a partial answer, by placing a special character in the digit(s) of the passcode that it is uncertain about.

### C.4.2 ANALYSIS

Let us assume for concreteness that each pair of adjacent chunks share an overlap of length $m/2$, and $m_i = m$ for all $i \in [k-1]$, while $m/2 \le m_k \le m$. In this case, we have $k = \lceil 2n/m - 1 \rceil$.

**Error metrics.** Our analysis of error metrics for this task is similar to that for the previous retrieval example. In particular, there are two possible failure modes in the retrieval step, and conclusions for the errors of the final solution returned by the overall algorithm are dependent on whether a Type-1 or Type-2 LLM is being used. For example, if a Type-1 LLM, which will not mistakenly retrieve irrelevant sentences from the input text, is used within the algorithm, then a smaller value of $m$ implies higher accuracy of retrieval in Step 2 of the algorithm, which further leads to lower error metrics for the solution returned by the final aggregation step.

**Cost metrics.** For simplicity, let us assume that a Type-1 LLM is used within the overall algorithm. Consequently, in Step 2 of the algorithm, each LLM call has $\mathcal{L}_{\text{pre}} \le \mathcal{L}_{\text{sys}} + O(m)$ and $\mathcal{L}_{\text{dec}} = O(1)$, since only the relevant text within the chunk is retrieved. Moreover, among the $k$ LLM calls, only $O(1)$ of them return answers that are not "None". By excluding the "None" results from the final aggregation step, the last LLM call has $\mathcal{L}_{\text{pre}} \le \mathcal{L}_{\text{sys}} + O(1)$ and $\mathcal{L}_{\text{dec}} = O(1)$. Putting things together, with a degree of parallelism $p$ and the number of chunks $k = \lceil 2n/m - 1 \rceil$, the cost metric $\mathcal{C}$ of the overall algorithm is bounded by

$$\mathcal{C} = \mathcal{C}(\text{sub-tasks}) + \mathcal{C}(\text{aggregation}), \quad \text{where}$$

$$\mathcal{C}(\text{sub-tasks}) \leq \left\lceil \frac{k}{p} \right\rceil \times \left( \mathcal{C}_{\text{pre}}(\mathcal{L}_{\text{sys}} + m) + \mathcal{C}_{\text{dec}}(\mathcal{L}_{\text{sys}} + m, 1) \right),$$

$$\mathcal{C}(\text{aggregation}) \leq \mathcal{C}_{\text{pre}}(\mathcal{L}_{\text{sys}} + 1) + \mathcal{C}_{\text{dec}}(\mathcal{L}_{\text{sys}} + 1, 1).$$

### C.4.3 EXPERIMENTS

We empirically validate the performance of the proposed LLM-based algorithm with a `Llama-3-70B` model. Two error metrics are considered: the exact-match error taking value in $\{0, 1\}$, and the fraction of incorrect digits, which takes value in $[0, 1]$ and is always no larger than the exact-match error.

In Figure 11a, we vary the problem size $n$, and set $m = n$ in the LLM-based algorithm. It is observed that the error metrics are monotonely increasing in $n$, and approach zero as $n$ decreases. Most cost metrics are also increasing in $n$, except for the number of decoding tokens, which is supposed to be determined by the number and lengths of the needles only, not the haystack, and thus should be insensitive to $n$.

In Figure 11b, we fix $n = 20000$ and vary the chunk size $m$. As was predicted by our analysis for a Type-1 LLM, a smaller value of $m$ implies lower error metrics, indicating the efficacy of task decomposition.

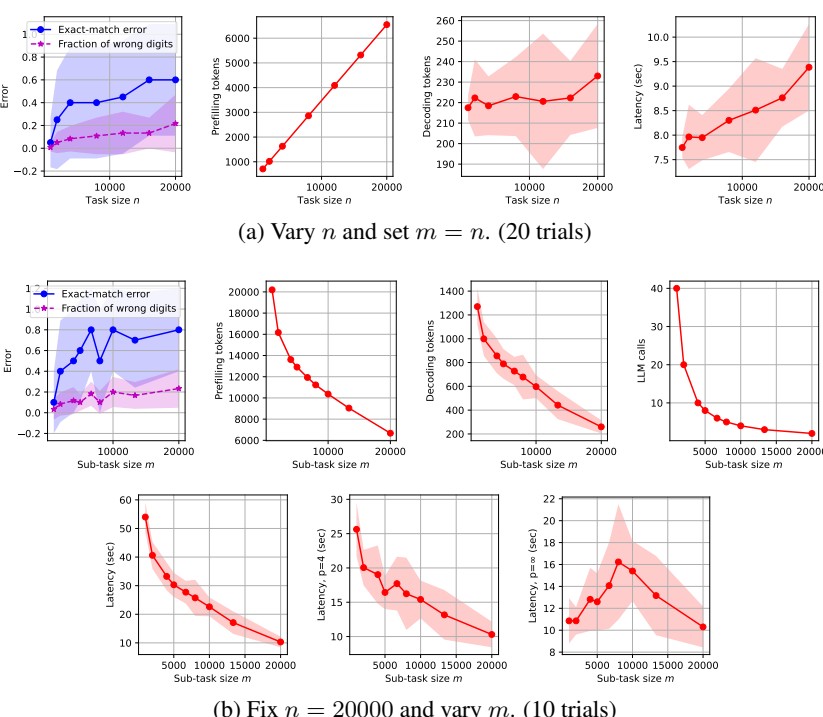

(a) Vary $n$ and set $m = n$. (20 trials)

(b) Fix $n = 20000$ and vary $m$. (10 trials)

Figure 11: Empirical results for RAG with `Llama-3-70B`.

## C.5 EXAMPLE: LONG-TEXT SUMMARIZATION WITH CHUNKING

For the final example of this section, we apply our analytical framework to the task of long-text summarization (Kryscinski et al., 2022; Chang et al., 2024), where generating a summary for the long input text with a single LLM call is infeasible. Algorithms that process the long text to be summarized by chunks have been proposed, such as hierarchical merging (Wu et al., 2021) and incremental updating (OpenAI, 2024b). The former first generates one summary for each chunk independently and then merge these intermediate summaries into the final one, while the latter maintains and updates a global summary as the chunks get processed one by one in order. Since objective and quantitative evaluation for the summarization task is known to be challenging (Chang et al., 2024) and remains an active research area, we focus our study on the cost metrics of both algorithms. Our analysis leads to interesting observations that are different from those in previous examples, primarily due to the various options of setting the number of generated tokens $\mathcal{L}_{\mathsf{dec}}$ for each LLM call within the algorithms.

### C.5.1 PROCESSING CHUNKS IN PARALLEL

**Algorithm.** Let us consider the following algorithm that processes the chunks in parallel, which is a simplification of hierarchical merging (Chang et al., 2024) and visualized in Figure 2a:

1. Divide the input text of length $n$ into $k$ chunks of length $m_1, m_2, \ldots, m_k$.

2. Summarize each chunk with one LLM call, independently and in parallel. For the $i$-th chunk, the LLM is prompted to generate a summary of length no larger than $s_i$.

3. Invoke one LLM call to merge the summaries into a single one of length no larger than $s$.

Note that the targeted lengths of the intermediate and final summaries, denoted by $\{s_i\}_{i \in [k]}$ and $s$, are hyperparameters of the algorithm that need to be pre-specified, in addition to the number of chunks $k$ and the chunk sizes $\{m_i\}_{i \in [k]}$.

**Analysis.** For notational convenience, we assume that $m_i \asymp m := n/k$ for all $i \in [k]$.

First, consider each individual LLM call: the cost of summarizing the $i$-th chunk can be bounded by $\mathcal{C}_{\mathsf{LLM}}(\mathcal{L}_{\mathsf{sys}} + m, s_i)$, while the cost of the final merging step can be bounded by $\mathcal{C}_{\mathsf{LLM}}(\mathcal{L}_{\mathsf{sys}} + \sum_{i \in [k]} s_i, s)$. To further simplify notations, we assume that $s_i = s_1$ for all $i \in [k]$, which will soon be justified. Then the total cost $\mathcal{C}$ of the overall algorithm can be bounded by

$$\mathcal{C} \le \mathcal{C}(\text{summarize chunks}) + \mathcal{C}(\text{merge summaries})$$
$$= k \times \mathcal{C}_{\mathsf{LLM}}(\mathcal{L}_{\mathsf{sys}} + m, s_1) + \mathcal{C}_{\mathsf{LLM}}(\mathcal{L}_{\mathsf{sys}} + k \times s_1, s),$$

while the end-to-end latency with parallelism degree $p$ can be bounded by

$$\mathcal{C} \le \lceil \frac{k}{p} \rceil \times \mathcal{C}_{\mathsf{LLM}}(\mathcal{L}_{\mathsf{sys}} + m, s_1) + \mathcal{C}_{\mathsf{LLM}}(\mathcal{L}_{\mathsf{sys}} + k \times s_1, s).$$

Let us specify the hyperparameters $s_1$ and more generally $\{s_i\}$, assuming that the targeted length $s$ of the final summary has been determined. If the targeted information that we wish to be included in the final summary is distributed evenly across the input text, e.g. in a story with a linear narrative, then it is reasonable to set $s_i = s_1 = s/k$ for all $i \in [k]$. In other scenarios where the targeted information is distributed unevenly (e.g. in query-focused summarization (Laban et al., 2024; Vig et al., 2022)), it is generally safer to set $s_i = s_1 = s$ for all $i \in [k]$. The latency with parallelism degree $p$ in both cases is summarized as follows:

$$\mathcal{C} \le \begin{cases} \lceil k/p \rceil \times \mathcal{C}_{\mathsf{LLM}}(\mathcal{L}_{\mathsf{sys}} + m, s/k) + \mathcal{C}_{\mathsf{LLM}}(\mathcal{L}_{\mathsf{sys}} + s, s) & \text{if} \quad s_1 = s/k, \\ \lceil k/p \rceil \times \mathcal{C}_{\mathsf{LLM}}(\mathcal{L}_{\mathsf{sys}} + m, s) + \mathcal{C}_{\mathsf{LLM}}(\mathcal{L}_{\mathsf{sys}} + k \times s, s) & \text{if} \quad s_1 = s. \end{cases}$$

Further analysis can be derived from here. To give an example, let us focus on the case of $s_1 = s$ and try to find the chunk size $m$ that minimizes the latency with parallelism degree $p$. For simplicity, we ignore the length $\mathcal{L}_{\mathsf{sys}}$ of the system prompt and the limit $\overline{m}$ on the context window size. We also

assume that the time complexity of each LLM call is linear, namely $\mathcal{C}_{\mathsf{LLM}}(\mathcal{L}_{\mathsf{pre}}, \mathcal{L}_{\mathsf{dec}}) \lesssim \mathcal{L}_{\mathsf{pre}} + \mathcal{L}_{\mathsf{dec}}$. Then we have

$$\mathcal{C} \leq \lceil \frac{k}{p} \rceil \times \mathcal{C}_{\mathsf{LLM}}(\mathcal{L}_{\mathsf{sys}} + m, s) + \mathcal{C}_{\mathsf{LLM}}(\mathcal{L}_{\mathsf{sys}} + k \times s, s) \lesssim \lceil \frac{k}{p} \rceil \times (m + s) + k \times s.$$

Even in such an oversimplified case, finding the optimal chunk size $m$ is non-trivial:

- For large $m \geq \widetilde{m} := n/p$ such that $k = n/m < p$, we have $\lceil k/p \rceil = 1$, and hence

$$\mathcal{C} \lesssim m + s + k \times s = m + \frac{n \times s}{m} + s.$$

  The right-hand side attains the minimum at $\widehat{m} := \sqrt{n \times s}$.

- For small $m < \widetilde{m} = n/p$, we have the approximation $\lceil k/p \rceil \approx k/p$, and hence

$$\mathcal{C} \lesssim \frac{k}{p} \times (m + s) + k \times s = k \times \left( \frac{m}{p} + \left( \frac{1}{p} + 1 \right) \times s \right)$$
$$\lesssim \frac{n}{m} \times \left( \frac{m}{p} + s \right) = n \times \left( \frac{1}{p} + \frac{s}{m} \right).$$

  The right-hand side is monotonely decreasing in $m$.

Given the above, it can be verified that the optimal chunk size $m^\star$ that minimizes the latency $\mathcal{C}$ can be estimated by $m^\star \asymp \max\{\widehat{m}, \widetilde{m}\} = \max\{\sqrt{n \times s}, n/p\}$.

### C.5.2 PROCESSING CHUNKS SEQUENTIALLY

**Algorithm.** Let us consider the following algorithm that processes the chunks sequentially, which is a simplification of incremental updating (Chang et al., 2024) and visualized in Figure 2b:

1. Divide the input text of length $n$ into $k$ chunks of length $m_1, m_2, \ldots, m_k$.

2. Initialize the global summary as an empty string, and update it incrementally via processing the chunks in order: at the $i$-th iteration, invoke one LLM call to utilize the global summary and the $i$-th chunk for generating a new global summary of length no larger than $s_i$.

3. The output of the overall algorithm is the global summary returned by the final LLM call.

**Analysis.** While this algorithm clearly has a sequential nature, our previous analysis for parallel decomposition can be easily adapted for this case. For the $i$-th step of updating the global summary, we have $\mathcal{L}_{\mathsf{pre}} \leq \mathcal{L}_{\mathsf{sys}} + m + s_{i-1}$ and $\mathcal{L}_{\mathsf{dec}} \leq s_i$, and thus its cost is bounded by

$$\mathcal{C}_i \leq \mathcal{C}_{\mathsf{LLM}}(\mathcal{L}_{\mathsf{sys}} + m + s_{i-1}, s_i).$$

Therefore, the total cost of the overall algorithm satisfies

$$\mathcal{C} \leq \sum_{i \in [k]} \mathcal{C}_i \leq \sum_{i \in [k]} \mathcal{C}_{\mathsf{LLM}}(\mathcal{L}_{\mathsf{sys}} + m + s_{i-1}, s_i).$$

Further analysis can provide insights for choosing the hyperparameters that minimize the total cost, though we omit the details here to avoid repetition.

# D  HIERARCHICAL DECOMPOSITION (EXTENDED)

This section studies the pattern of *hierarchical decomposition* for LLM-based algorithms, where the original task is decomposed into multiple sub-tasks, and each of them can be further decomposed into more lower-level sub-tasks. This pattern is strictly more expressive than parallel decomposition studied in the previous section, and hence capable of solving more problems.

For concreteness, we investigate a challenging version of the RAG task studied in Appendix C.4 that requires multi-hop reasoning (Yang et al., 2018; Li et al., 2024a). Recall that in the previous RAG example, given a question and multiple text chunks, the algorithm following the pattern of parallel decomposition will first retrieve useful sentences from each chunk, and then invoke one LLM call to answer the question based on the retrieved information. Such a method might not be feasible in more challenging scenarios, where the needles embedded in the haystack are logically related, and some of the needles are related to the targeted question only *indirectly* via connection to other needles. For example, suppose that the question is "What is the numeric value of A?", while the needles are "A = B", "B = C", and "C = 100", located separately in different chunks. Retrieving needles from their corresponding chunks solely based on the targeted question will certainly fail in this case.

To tackle this challenge, a natural idea is to extend the RAG algorithm considered in Appendix C.4, allowing it to perform multiple rounds of iterative retrieval and reasoning. For each round, the algorithm performs retrieval based on the targeted question as well as additional information from previous rounds, followed by reasoning about the retrieved sentences and deciding whether the targeted question can be answered, or further retrieval and reasoning is needed. Similar approaches have been widely adopted in prior work, e.g. in the Selection-Inference framework (Creswell et al., 2023), in a RAG system with iterative follow-up questions for applications in medicine (Xiong et al., 2024), or as a technique of extending the effective context window of a LLM-powered agent system (Qwen-Team, 2024). The resulting algorithm for reasoning with iterative retrieval exhibits a hierarchical structure: the original task is decomposed into multiple sequential rounds, and each round is further decomposed into multiple steps of retrieval and reasoning.

In the rest of this section, we first elaborate the concrete setup and algorithms for this case study, and then demonstrate our analysis of accuracy and efficiency based on the proposed analytical framework, as well as insights derived from it. Finally, numerical experiments are conducted to validate the efficacy and scalability of the considered algorithm, as well as our analysis and insights.

## D.1  CONCRETE SETUP

For concreteness, let us consider the task of finding the numeric value of a targeted variable embedded within a grid, similar to the setting considered by Ye et al. (2024). A visualization can be found in Figure 12. Configurations of the grid include its depth $d$, width $w$, and degree $g$, which control the difficulty of this task. More specifically, the grid consists of $d$ levels, each containing $w$ variables; each variable at a certain level is a function (e.g. addition, substraction, maximum or minimum) of $g$ variables at the next level, except for the leaf variables at the final level, whose numeric values are given. Such information is provided in clues of the form "A2 = B1 + B3" or "C1 = 10", each of which corresponds to one variable. As a result, the total number of clues is $d \times w$, and each clue has length $O(g)$. We refer to the clues that are necessary and sufficient for calculating the targeted variable as the needles, which constitute a directed acyclic graph (DAG) embedded within the grid. It is assumed that the level of reasoning required in this case study is non-trivial yet also simple enough, in the sense that one single LLM call with step-by-step reasoning is sufficient for answering the targeted question correctly if all needles are given *a priori*.

## D.2  ALGORITHM

To solve this task, we consider an LLM-based algorithm that involves multiple rounds of retrieval and reasoning, which is visualized in Figure 13 and explained in the following:

1. Divide the input clues into $k$ chunks. In addition, initialize an empty list of references, i.e. clues retrieved by LLM calls, which will be maintained and updated throughout the algorithm.

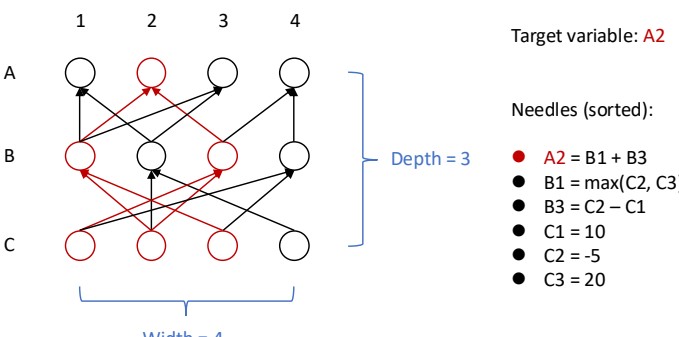

Figure 12: An example grid of variables with depth $d = 3$, width $w = 4$, and degree $g = 2$. The targeted variable is A2 at the top level, which is a function of B1 and B3 at the next level. The variables whose numeric values are necessary and sufficient for calculating A2 are highlighted in red, and their corresponding clues constitute a DAG of needles.

2. For each round of the algorithm, invoke $k$ sequential (Figure 13a) or parallel (Figure 13b) LLM calls for retrieval from $k$ chunks based on the targeted question and references, then invoke one LLM call to reason about the updated references and try to answer the targeted question.

   - If the LLM returns an answer, then the overall algorithm outputs that answer and terminate.
   - If the LLM cannot answer, and no additional clue has been retrieved in this round, then the algorithm terminates and fails[6].
   - Otherwise, move on to the next round of retrieval and reasoning, with the updated references.

**Options for reasoning and answering.** We consider two options of prompting the LLM to reason about the references and answer the targeted question: answering directly, or thinking step by step before answering (Kojima et al., 2022). It is reasonable to expect that the latter will boost the accuracy while incurring higher costs, which will soon be investigated analytically and quantitatively in Appendix D.3.

**Options for retrieval.** We consider two options, referred to as "cyclic" and "parallel", for retrieval from multiple chunks during each round of the algorithm.

   - With the "cyclic" option (Figure 13a), chunks are processed sequentially; more specifically, after retrieval for each chunk, the retrieved clues are added to the list of references immediately, before retrieval for the next chunk starts. Consequently, the chunks are processed in a cyclic manner throughout the overall algorithm, which gives rise to the name of this option.
   - With the "parallel" option (Figure 13b), chunks are processed independently and in parallel, after which the list of references is updated once. This exemplifies using parallel decomposition (cf. Section 3) as a building block for more complicated LLM-based algorithms.

With the "cyclic" option, the overall algorithm typically need fewer rounds to answer the targeted question correctly, since each LLM call for retrieval always leverages the most updated references. On the other hand, the "parallel" option can leverage parallelism of LLM calls for reducing the end-to-end latency of the overall algorithm. The comparison between these two options will soon be elaborated in Appendix D.3.

---

[6]This is due to the assumption that greedy decoding, which is deterministic, is adopted. In this case, running one more round of retrieval and reasoning will give the same result as that of the current round, which will be useless and wasteful. A potential improvement is to adaptively decrease (e.g. halve) the chunk size and continue the algorithm with more rounds; such improvements are not the focus of this work, and hence omitted here.

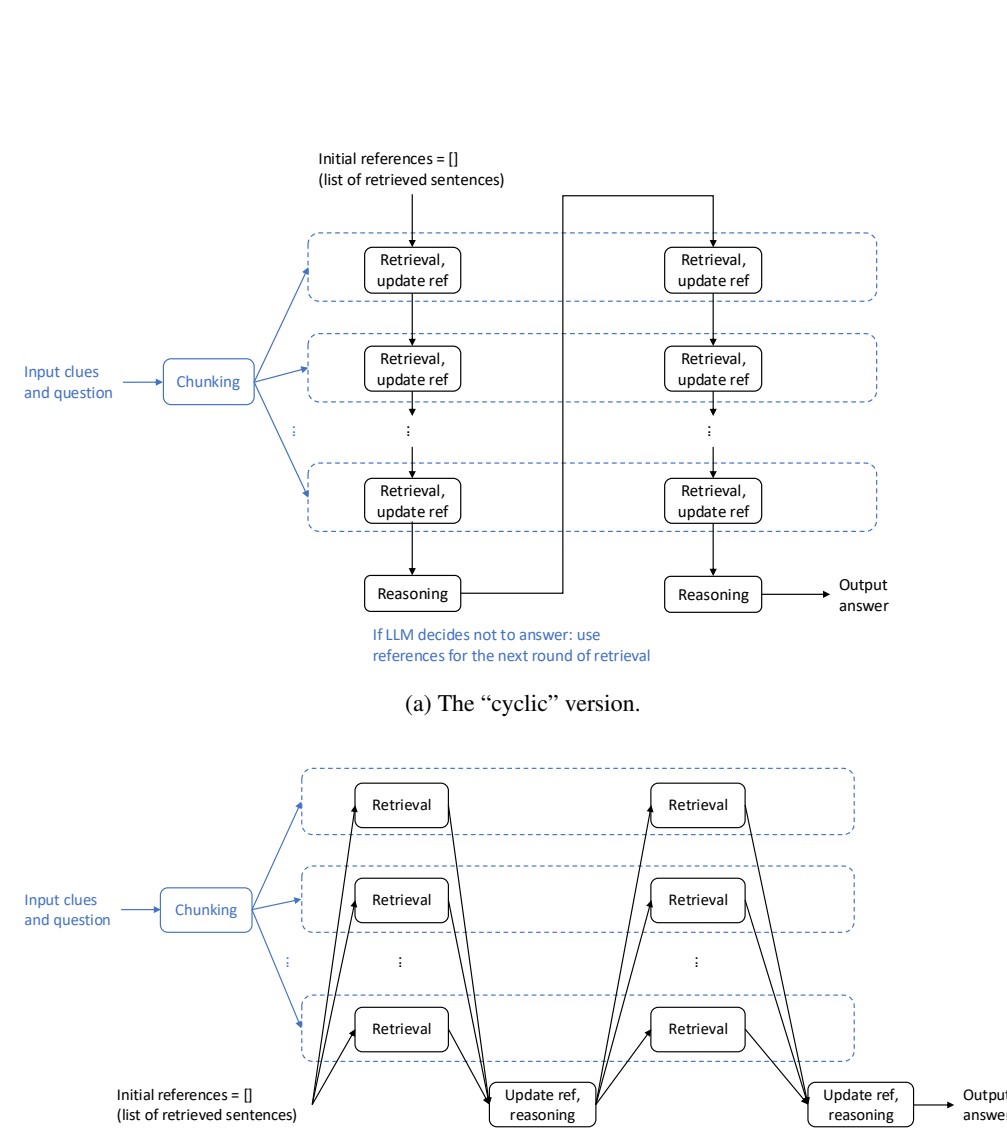

(a) The "cyclic" version.

(b) The "parallel" version.

Figure 13: Algorithms for reasoning with iterative retrieval, which exhibit the pattern of hierarchical decomposition. While the number of rounds is assumed to be 2 in this visualization, it is actually determined adaptively by the algorithm itself at runtime. For clarity of visualization, some LLM or non-LLM nodes (cf. Figure 1) are merged into one, and an arrow from the "chunking" node to a dashed block means that each "retrieval" node within the dashed block takes the corresponding chunk as input.

*Remark* 2. The algorithm under consideration exhibits the pattern of hierarchical decomposition, in that the overall task is decomposed into a sequence of rounds, and each round is further decomposed into multiple steps of retrieval and reasoning. Another feature of this algorithm is that its computational graph is constructed dynamically at runtime, since the number of rounds is determined adaptively by the algorithm itself.

### D.3 ANALYSIS

#### D.3.1 ERROR METRICS

There are two major failure modes for this algorithm of reasoning with iterative retrieval:

- *Mistakes in reasoning and answering.* For example, the LLM call responsible for reasoning and answering might mistakenly decide that it is not yet ready to answer the targeted question, or make mistakes when doing the arithmetic and thus return a wrong numeric value, even when all needles have been successfully retrieved. It is also possible that the LLM call mistakenly returns a numeric value (e.g. the value of a variable whose name is similar to that of the targeted variable), while the needles have not been fully retrieved yet. The probability of this failure mode is particularly related to whether the LLM call responsible for reasoning and answering is prompted to answer directly or think step by step before answering.

- *Failure to retrieve all needles.* As long as the ground-truth DAG of needles has a limited size, the probability of this failure mode is largely determined by the choice of the chunk size. Since this has been thoroughly investigated in our previous retrieval (Appendix C.3) and RAG (Appendix C.4) examples, it will not be our focus in this section.

#### D.3.2 COST METRICS

**Notations.** Recall from Eq. (2) that $\mathcal{C}_{\mathsf{LLM}}(\mathcal{L}_{\mathsf{pre}}, \mathcal{L}_{\mathsf{dec}}) = \mathcal{C}_{\mathsf{pre}}(\mathcal{L}_{\mathsf{pre}}) + \mathcal{C}_{\mathsf{dec}}(\mathcal{L}_{\mathsf{pre}}, \mathcal{L}_{\mathsf{dec}})$ stands for the cost of one LLM call with a prompt of length $\mathcal{L}_{\mathsf{pre}}$ and generated text of length $\mathcal{L}_{\mathsf{dec}}$. We also adopt some notations from Section 3.1 in our previous study of parallel decomposition: $n$ represents the total length (in tokens or characters) of the input clues, $m$ represents an upper bound for the chunk size, $k \lesssim n/m$ denotes the number of chunks, and $\mathcal{L}_{\mathsf{sys}}$ denotes an upper bound for the length of the system prompt for each LLM call.

In addition, let $r$ denote the number of rounds, which is determined adaptively by the algorithm itself at runtime. Finally, let $\ell$ be the total length of the needles, i.e. clues that are necessary and sufficient for answering the targeted question correctly. The value of $\ell$ can depend on the depth $d$, width $w$ and degree $g$ of the grid in various ways:

- If $g = 1$, then the needles form a chain of length $d$, in the form of "A = B", "B = C", "C = D", and so on. In this case, we have $\ell \lesssim d$.

- If the width $w > d^{g-1}$ is sufficiently large, then the number of needles can be upper bounded by the size of a tree with $d$ levels and $g$ children per node, namely $1 + g + g^2 + \cdots + g^{d-1} = (g^d - 1)/(g - 1)$. Since each needle has length $O(g)$, we have $\ell \lesssim g \times (g^d - 1)/(g - 1)$.

- If the width $w \ll d^{g-1}$ is limited, then a tighter bound for the number of needles will be $d \times w$, and thus $\ell \lesssim g \times d \times w$.

**The "cyclic" version (Figure 13a).** Let us first study the cost of each LLM call.

- An LLM call responsible for retrieval takes one chunk and the references as input, which has length $O(m + \ell)$. In addition, it is supposed to output a list of clues, which has length $O(\ell)$, that is relevant to the targeted question. In other words, we have $\mathcal{L}_{\mathsf{pre}} \leq \mathcal{L}_{\mathsf{sys}} + O(m + \ell)$, $\mathcal{L}_{\mathsf{dec}} \lesssim \ell$, and thus

$$\mathcal{C}(\text{retrieval}) \leq \mathcal{C}_{\mathsf{LLM}}(\mathcal{L}_{\mathsf{sys}} + m + \ell, \ell)$$

- An LLM call responsible for reasoning and answering takes the references of length $O(\ell)$ as input, namely $\mathcal{L}_{\mathsf{pre}} \leq \mathcal{L}_{\mathsf{sys}} + O(\ell)$. As for the length of generated text $\mathcal{L}_{\mathsf{dec}}$, it is reasonable

to assume that $\mathcal{L}_{\mathsf{dec}} \lesssim 1$ if the LLM is prompted to answer directly, and $\mathcal{L}_{\mathsf{dec}} \lesssim \ell$ if it is prompted to do the calculation step by step before returning its final answer[7]. Consequently, one has

$$\mathcal{C}(\text{reasoning}) \leq \mathcal{C}_{\mathsf{LLM}}(\mathcal{L}_{\mathsf{sys}} + \ell, 1 \text{ or } \ell).$$

Now we are ready to find out the cost of the overall algorithm with $r$ rounds of iterative retrieval (for $k$ chunks) and reasoning:

$$\mathcal{C} \leq r \times \mathcal{C}(\text{one round})$$

$$= r \times \Big( k \times \mathcal{C}(\text{retrieval}) + \mathcal{C}(\text{reasoning}) \Big)$$

$$\leq r \times \Big( k \times \mathcal{C}_{\mathsf{LLM}}(\mathcal{L}_{\mathsf{sys}} + m + \ell, \ell) + \mathcal{C}_{\mathsf{LLM}}(\mathcal{L}_{\mathsf{sys}} + \ell, 1 \text{ or } \ell) \Big).$$

In particular, this bound quantifies how the cost of LLM calls responsible for reasoning and answering, namely $r \times \mathcal{C}_{\mathsf{LLM}}(\mathcal{L}_{\mathsf{sys}} + \ell, 1 \text{ or } \ell)$, only occupies a small fraction of the total cost of the overall algorithm.

**The "parallel" version (Figure 13b).** Analysis of cost metrics for the "parallel" version is largely the same as that for the "cyclic" version, with the following two major differences:

- For the same task instance, the "parallel" version requires the same or a larger number of rounds, since in the "cyclic" version, each retrieval step always leverages the most updated references. For example, consider the case with degree $g = 1$ and a chain of $d$ needles located separately in different chunks. Then, the number of rounds will be $r = d$ for the "parallel" version, but can be as small as $r = 1$ for the "cyclic" version, if the needles happen to appear in the right order in the original clues.

- If the cost metric of interest is the end-to-end latency with parallelism degree $p$, then the $k$ LLM calls for retrieval within each round can be parallelized, which implies

$$\mathcal{C} \leq r \times \left( \lceil \frac{k}{p} \rceil \times \mathcal{C}(\text{retrieval}) + \mathcal{C}(\text{reasoning}) \right)$$

$$\leq r \times \left( \lceil \frac{k}{p} \rceil \times \mathcal{C}_{\mathsf{LLM}}(\mathcal{L}_{\mathsf{sys}} + m + \ell, \ell) + \mathcal{C}_{\mathsf{LLM}}(\mathcal{L}_{\mathsf{sys}} + \ell, 1 \text{ or } \ell) \right). \qquad (7)$$

As a result, the end-to-end latency of the "parallel" version with a sufficiently large parallelism degree $p$ can be potentially smaller than that of the "cyclic" version.

### D.3.3 INSIGHTS

In sum, we derive two major insights from the above analysis of error and cost metrics:

1. LLM nodes responsible for reasoning and answering only occupy a small fraction of the costs of the overall algorithm, while playing a critical role in the output of the algorithm and thus in the error metrics. Therefore, our general recommendation is to prompt these LLM calls to think step by step before answering, instead of answering directly, which will most likely boost the accuracy significantly with negligible loss in efficiency.

2. Regarding the "cyclic" and "parallel" options for retrieval, we conclude that each of them has its own pros and cons. With the "cyclic" option, fewer rounds of retrieval and reasoning are needed (thanks to the timely updates to the references), but the downside is that all retrieval steps have to be executed sequentially. In contrast, with the "parallel" option, the algorithm typically requires more rounds of retrieval and reasoning and thus larger costs in terms of most metrics, but can leverage parallelism of LLM calls for reducing the end-to-end latency. Therefore, whether the "cyclic" or "parallel" option is preferred largely depends on which cost metric is of more concern.

---

[7]There might exist more precise characterizations of $\mathcal{L}_{\mathsf{dec}}$ in this case, e.g. $\mathcal{L}_{\mathsf{dec}} \lesssim \ell \times g$ if the LLM tends to take $g$ steps for calculating the sum of $g$ numbers. For simplicity, we mostly assume $g = O(1)$ in this case study, and thus stick to the assumption that $\mathcal{L}_{\mathsf{dec}} \lesssim \ell$.

## D.4 EXPERIMENTS

We validate our analysis via experiments with a `Qwen2-72B-Instruct` model (Yang et al., 2024), supported by vLLM (Kwon et al., 2023) and running on a server with 4 Nvidia A100-80G GPUs. For each experiment, we vary the depth, width or degree of the grid of variables (which controls the difficulty of task instances), while validating Insight 1 by comparing two options for prompting LLM calls that are responsible for reasoning and answering, or Insight 2 by comparing the "cyclic" and "parallel" options for retrieval.

Error metrics of interest include (1) the exact-match error, which takes value 0 if the answer returned by the algorithm matches the ground-truth solution exactly, and value 1 otherwise; (2) the absolute error, i.e. the absolute value of the difference between the algorithm's answer and the ground-truth solution; and (3) the missed-coverage error, i.e. the ratio of needles that the algorithm fails to retrieve and hence are not included in the references maintained by the algorithm. Note that missed coverage might arise not just from failures of the LLM calls responsible for retrieval, but also from the possibility that the algorithm terminates too early with fewer rounds than necessary, due to hallucination by the LLM calls responsible for reasoning and answering. Cost metrics of interest are the same as those explained in Appendix C, plus the number of rounds, a metric specific to the hierarchical pattern of algorithms considered in this section.

*Remark* 3 (Mitigating hallucination in retrieval). During our early experiments, we observed that LLMs tend to make up clues not present in the input during the retrieval steps. To address this, we prompt the LLM to return exact copies of clues from the original text, and further use a symbolic program to check whether the retrieved clues are indeed exact copies (which, from the perspective of LLM-powered agent systems, might be regarded as tool use). Only those passing the test will be added to the list of references. It has been confirmed empirically that this simple method effectively mitigates the errors caused by hallucination of LLMs during retrieval, making the overall algorithm much more robust and accurate in our experiments.

**Comparing two prompting options.** For this experiment, we consider shallow and wide grids of variables, with depth $d = 2$ and width $w = 300$. The degree $g$ varies, controlling the difficulty in arithmetic. The algorithm uses a fixed chunk size of 50 clues, and the "parallel" version of retrieval. In this case, the ideal number of rounds should be 2 if each retrieval step succeeds.

Empirical results for this experiment are illustrated in Figure 14, which confirm Insight 1, i.e. prompting the LLM to think step by step for reasoning and answering significantly boosts the accuracy of the overall algorithm while incurring minor overhead in cost metrics, compared to answering directly. Note from the last row, though, that the relative difference in terms of end-to-end latencies increases with the parallelism degree $p$, which has been predicted by our analysis as well, in particular Eq. (7).

**Comparing the "cyclic" and "parallel" versions.** In the following experiments, we consider grids with degree $g = 1$ and thus chains of needles. We either fix the depth $d = 5$ and vary the width $w$, or fix the width $w = 100$ and vary the depth $d$. The algorithm uses a fixed chunk size of 100 clues, and prompts the LLM calls responsible for reasoning and answering to think step by step.

Empirical results for these two experiments can be found in Figures 15 and 16, which confirm that the considered algorithm achieves high accuracy for a wide range of task configurations. Insight 2 from our analysis is also validated: compared with the "cyclic" version, the "parallel" version incurs a larger number of rounds and hence higher cost metrics, except for the end-to-end latencies with parallelism, for which the benefits of parallelism outweight the downside of requiring more rounds.

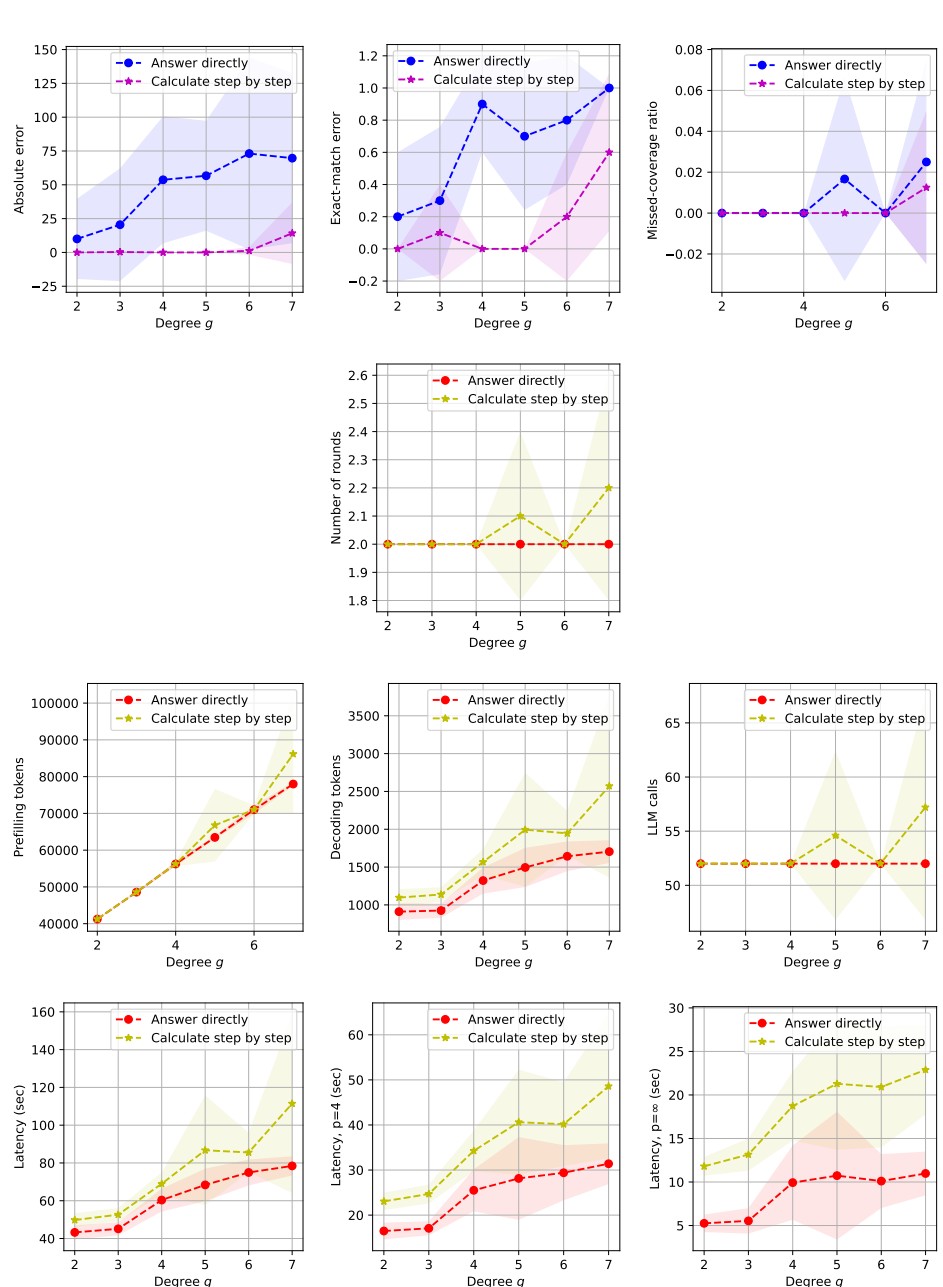

Figure 14: Empirical results (10 trials) for reasoning with iterative retrieval, where two options of prompting the LLM calls responsible for reasoning and answering are compared. The depth $d = 2$ and width $w = 300$ are fixed, while the degree $g$ varies. The algorithm uses a chunk size of 50 clues, and the "parallel" version of retrieval.

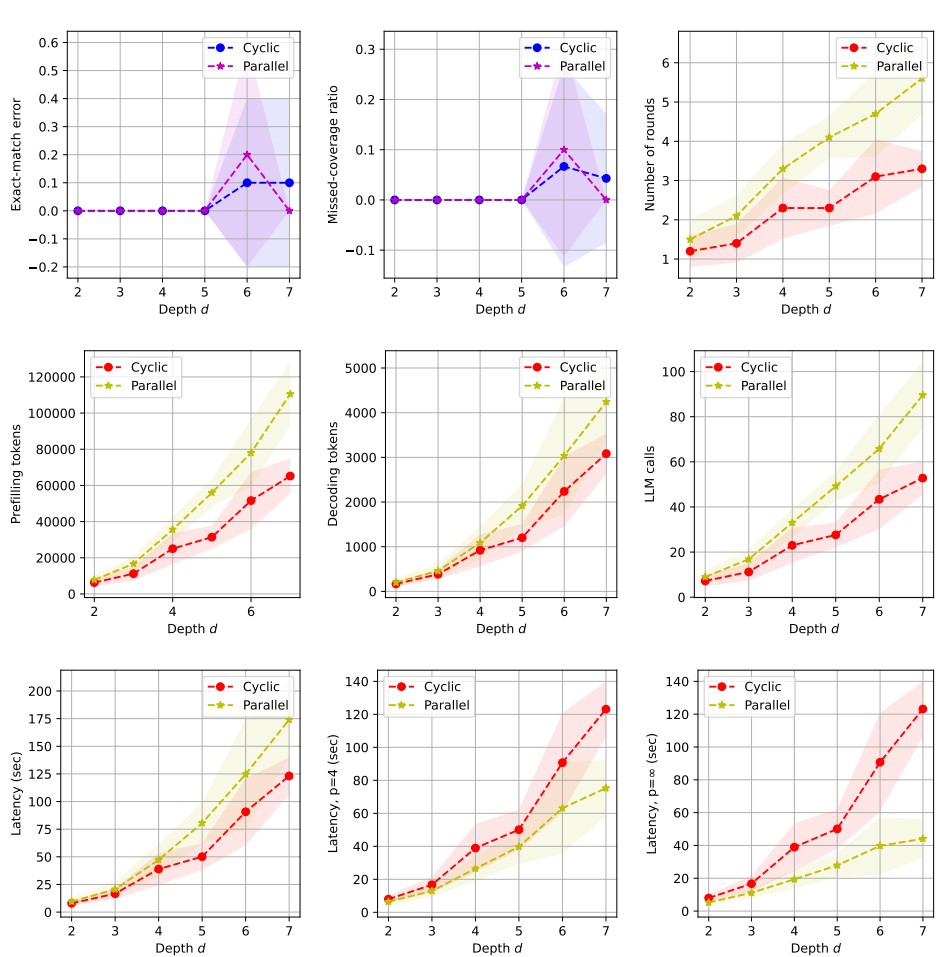

Figure 15: Empirical results (10 trials) for reasoning with iterative retrieval, where two options of retrieval, namely "cyclic" and "parallel", are compared. The degree $g = 1$ and width $w = 100$ are fixed, while the depth $d$ varies. The algorithm uses a chunk size of 100 clues, and prompts the LLM calls responsible for reasoning and answering to think step by step.

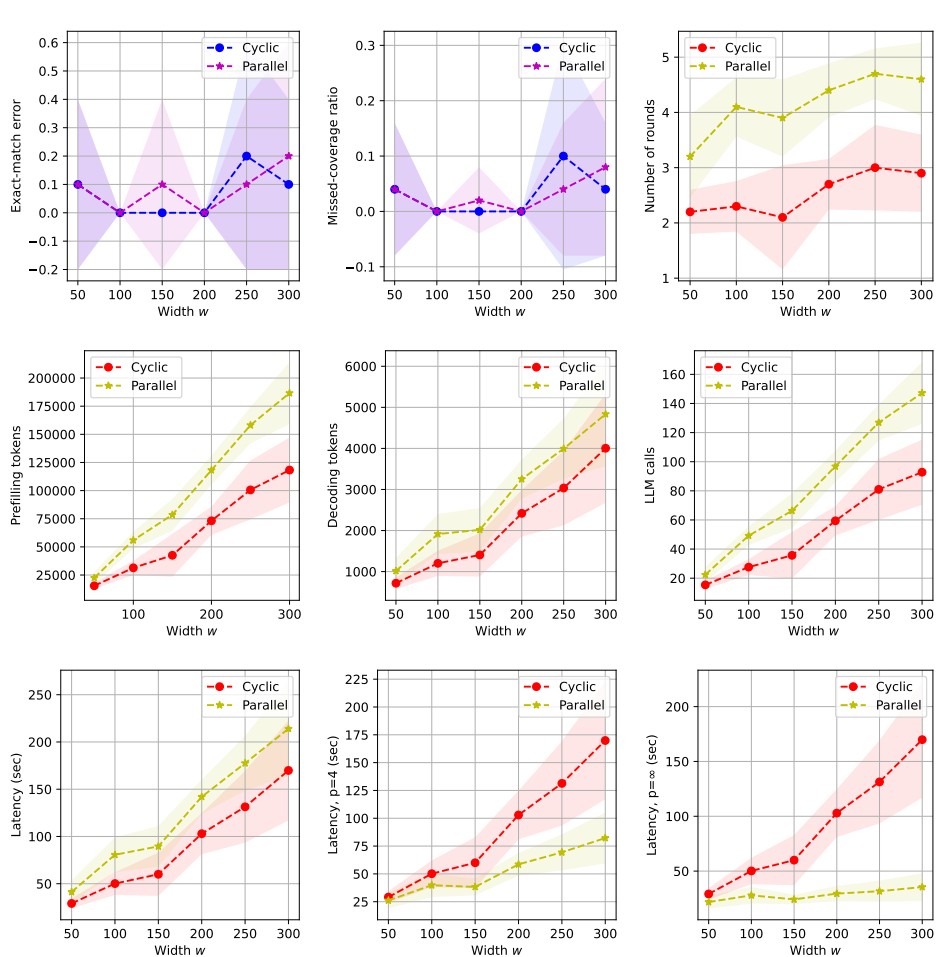

Figure 16: Empirical results (10 trials) for reasoning with iterative retrieval, where two options of retrieval, namely "cyclic" and "parallel", are compared. The degree $g = 1$ and depth $d = 5$ are fixed, while the width $w$ varies. The algorithm uses a chunk size of $100$ clues, and prompts the LLM calls responsible for reasoning and answering to think step by step.

# E    RECURSIVE DECOMPOSITION (EXTENDED)

This section studies LLM-based algorithms with *recursive task decomposition*, a pattern that is vastly different from those studied in previous sections, and yet still covered by the analytical framework proposed in Section 2. Roughly speaking, a recursive LLM-based algorithm starts from the original task of concern and recursively generates more intermediate sub-tasks, so that each sub-task can be solved by aggregating the solutions to its children tasks, while being unaware of other sub-tasks involved in the overall algorithm. In particular, solving and/or decomposing each sub-task can be achieved by LLM calls, while the outline of recursive task decomposition remains symbolic. Such LLM-based algorithms have been widely applied, e.g. in LAMBADA (Kazemi et al., 2023), LLM programs (Schlag et al., 2023), ADaPT (Prasad et al., 2024), THREAD (Schroeder et al., 2024), Recursion of Thought (Lee & Kim, 2023), Decomposed Prompting (Khot et al., 2023), ReDel (Zhu et al., 2024), among many others.

In the following, we first introduce the concrete task under consideration, and then design a recursive LLM-based algorithm for solving it, highlighting the dynamic construction of its computational graph at runtime. We then provide analysis for its accuracy and efficiency using the proposed framework, which is further validated with numerical experiments; along the way, we derive a formal guarantee for generic LLM-based algorithms, under the technical assumption of additive errors and bounded sensitivity.

## E.1    CONCRETE SETUP

For concreteness, we consider the same task introduced in Appendix D.1 and visualized in Figure 12, which is about calculating a targeted variable based on clues about a grid of many variables.

One major difference here is that we consider much larger depth $d$, width $w$ and/or degree $g$ for the grid of variables, which implies a larger DAG of needles, i.e. clues relevant to the targeted question. Even if all relevant clues are given *a priori*, the complex reasoning required to answer the targeted question correctly can be well beyond the capability of one single LLM call, which motivates decomposing the reasoning process into multiple LLM calls.

The other major difference is, we assume that the clues are not directly given in plain text, but rather need to be accessed via querying a database. For each query, the database takes a name as input, and returns a clue for the variable of the same name, e.g. "A2 = B1 + B3" for input "A2" or "C1 = 10" for input "C1". The LLM-based algorithm need to decide by itself what to query from the database. Such a setting is motivated by, and can be regarded as an abstraction of, practical scenarios where an autonomous agent in the wild need to actively retrieve relevant information by itself in order to accomplish a task (unlike in typical problem-solving benchmarks where such information is given), via querying a real database or knowledge graph, using a search engine, retrieving from documents, etc. In this case study, we assume that querying the database does not involve LLM calls, and thus neglect its costs in our analysis.

## E.2    ALGORITHM

We consider the following recursive LLM-based algorithm for solving this task. The overall algorithm maintains a dictionary of variables that have been calculated throughout its execution, and eventually outputs the value returned by applying a function named `ProcessNode` to the targeted variable.

The function `ProcessNode`, which takes the name of a variable as input and returns its numeric value, is defined in a recursive manner:

1. If the input variable is found in the dictionary of calculated variables, then return its value directly.

2. Otherwise, query the database with the input variable, and invoke one LLM call with the returned clue, which is prompted to either answer with a numeric value or conduct further task decomposition, i.e. identifying variables whose values will be necessary and sufficient for calculating the input variable.

   (a) If the LLM chooses to answer with a numeric value, then return it.

(b) Otherwise, invoke `ProcessNode` once for each variable identified by the LLM call, collect the answers returned by these children tasks, and finally invoke one LLM call to calculate and return the numeric value of the input variable.

We make a few comments about this recursive algorithm. (1) The computational graph of this algorithm, or a generic LLM-based algorithm following the pattern of recursive decomposition, is constructed dynamically in a depth-first-search style at runtime. See Figure 4 for a visualization. The final graph is symmetric about the nodes corresponding to the leaf variables (D, E); on the left are LLM calls for identifying children variables for the non-leaf variables (A, B, C), while on the right are LLM calls for calculating their numeric values. (2) The process of complex reasoning is decomposed recursively, so that each LLM call within this algorithm only need to handle a small step of reasoning, whose complexity is irrelevant to that of the overall task. (3) One particular configuration of this algorithm, like in the previous case study, is about how the LLM calls (especially the ones responsible for calculation) are prompted, e.g. to answer directly or think step by step. The impact of this design choice on the accuracy and efficiency of the overall algorithm will be investigated analytically and empirically soon in this section.

### E.3 ANALYSIS

#### E.3.1 ERROR METRICS

Recall from Eq. (1) that for a specific node $v$ with inputs $x_1, \ldots, x_k$, the error of its output $y$ is assumed to be upper bounded by $\mathcal{E}(y) \leq f_v(\mathcal{E}(x_1), \mathcal{E}(x_2), \ldots, \mathcal{E}(x_k))$ for some function $f_v$. For the concrete setting of the current case study, there are two major failure modes for the aforementioned algorithm: errors in recursive task decomposition, and errors in calculating the numeric values of specific variables. Empirical results during our early exploration suggested that LLMs make very few mistakes of the first kind, while the second failure mode is much more common, due to the limited arithmetic capabilities of current LLMs. Moreover, all mathematical operations considered in this concrete settings (including addition, substraction, maximum and minimum) are 1-Lipschitz continuous with respect to each input variable.

All these motivate us to derive the following error bound, which indeed holds true for any generic LLM-based algorithm represented by a directed acyclic graph (DAG), under the assumption of *additive errors and bounded sensitivity*. The proof is deferred to Section E.5 after the experiments.

**Proposition 3.** *Suppose that the assumption of additive errors and bounded sensitivity holds true for an LLM-based algorithm represented by a DAG, namely for each node $v$ with $k$ inputs $x_1, \ldots, x_k$ and a single output $y$, it holds that*

$$\mathcal{E}(y) \leq f_v\big(\mathcal{E}(x_1), \mathcal{E}(x_2), \ldots, \mathcal{E}(x_k)\big) := \mathcal{E}_v + S \times \sum_{i \in [k]} \mathcal{E}(x_i)$$

*for some node-specific additive error $\mathcal{E}_v$ and finite sensitivity parameter $S \geq 0$. Then the error of the output $y(v)$ of any node $v$, including the one that generates the final output of the overall algorithm, is bounded by*

$$\mathcal{E}(y(v)) \leq \sum_{w \in DAG} \sum_{path \in \mathcal{P}(w \to v)} S^{|path|} \times \mathcal{E}_w. \tag{8}$$

*Here, $|path|$ denotes the length of a path on the DAG, while $\mathcal{P}(w \to v)$ represents the set of paths from node $w$ to node $v$ if $w \neq v$, or a singleton set containing one hypothetical path of length $0$ if $w = v$. For the special case of $S = 1$, this upper bound can be simplified as $\mathcal{E}(y(v)) \leq \sum_{w \in DAG} |\mathcal{P}(w \to v)| \times \mathcal{E}_w$.*

This proposition precisely characterizes how the error $\mathcal{E}_w$ of each node $w$ impacts the accuracy of the overall algorithm, via the number and lengths of the paths from each node to the final output.

#### E.3.2 COST METRICS

Let us first consider the cost of each LLM call. For each leaf variable, one LLM call is needed for finding its numeric value based on its corresponding clue of length $O(1)$, which implies

$\mathcal{L}_{\text{pre}} \leq \mathcal{L}_{\text{sys}} + O(1)$ and $\mathcal{L}_{\text{dec}} \leq O(1)$, and thus $\mathcal{C}(\text{leaf}) \leq \mathcal{C}_{\text{LLM}}(\mathcal{L}_{\text{sys}} + 1, 1)$. For each non-leaf variable, two LLM calls are needed. One of them is responsible for task decomposition, i.e. identifying its $g$ children variables, which implies $\mathcal{L}_{\text{pre}} \leq \mathcal{L}_{\text{sys}} + O(g)$ and $\mathcal{L}_{\text{dec}} \leq O(g)$, and thus $\mathcal{C}(\text{decomposition}) \leq \mathcal{C}_{\text{LLM}}(\mathcal{L}_{\text{sys}} + g, g)$. The other LLM call is responsible for calculating the numeric value of the current variable based on the values of its children variables, which has $\mathcal{L}_{\text{pre}} \leq \mathcal{L}_{\text{sys}} + O(g)$, and $\mathcal{L}_{\text{dec}} \leq O(1)$ if the LLM is prompted to answer directly, or $\mathcal{L}_{\text{dec}} \leq \mathcal{L}_{\text{calc}}(g)$ for some function $\mathcal{L}_{\text{calc}}$ if it is prompted to do the calculation step by step (e.g. $\mathcal{L}_{\text{calc}}(g) \leq O(g)$ or $O(g^2)$, depending on how the LLM executes the calculation with $g$ variables). In sum, we have $\mathcal{C}(\text{calculation}) \leq \mathcal{C}_{\text{LLM}}(\mathcal{L}_{\text{sys}} + g, 1 \text{ or } \mathcal{L}_{\text{calc}}(g))$ for each LLM call responsible for calculation.

The analysis above has also revealed the total number of LLM calls. Let $n$ and $n_{\text{leaf}}$ be the number of relevant variables and the number of relevant leaf variables, respectively. These parameters are determined by $d, w, g$ and other factors, as is the case for the $\ell$ parameter (total length of relevant clues) in Appendix D. Recall from above that each leaf variable requires one LLM call for finding its value from its corresponding clue, while each non-leaf variable requires two LLM calls, one for task decomposition and one for calculation. Assuming that task decomposition is done correctly by the LLM for each non-leaf variable, the total number of LLM calls will be $2 \times (n - n_{\text{leaf}}) + n_{\text{leaf}} = 2 \times n - n_{\text{leaf}}$.

Putting things together, we finally arrive at the total cost of the overall recursive algorithm:

$$\mathcal{C} \leq (n - n_{\text{leaf}}) \times \Big( \mathcal{C}_{\text{LLM}}\big(\mathcal{L}_{\text{sys}} + g, g\big) + \mathcal{C}_{\text{LLM}}\big(\mathcal{L}_{\text{sys}} + g, 1 \text{ or } \mathcal{L}_{\text{calc}}(g)\big)\Big) + n_{\text{leaf}} \times \mathcal{C}_{\text{LLM}}(\mathcal{L}_{\text{sys}}, 1). \tag{9}$$

In particular, prompting the LLM to do the calculation step by step, which is anticipated to significantly improve the accuracy of the overall algorithm, will increase most cost metrics (though not for the number of LLM calls or the total number of prefilling tokens) by a *multiplicative* factor, in contrast to the previous case study in Appendix D where step-by-step prompting incurs a minor *additive* cost.

### E.4 EXPERIMENTS

We validate our analysis via experiments with a `Qwen2-72B-Instruct` model (Yang et al., 2024), supported by vLLM (Kwon et al., 2023) and running on a server with 4 Nvidia A100-80G GPUs. For each experiment, we vary the difficulty of the task through the depth $d$ or degree $g$ of the grid of variables, while comparing two options of prompting the LLM calls responsible for calculation, namely "answer directly" and "calculate step by step".

Empirical results are shown in Figures 17 and 18. For the former, we fix the width $w = 6$ and degree $g = 4$, while varying the depth $d$; for the latter, we fix the depth $d = 3$ and width $w = 100$, while varying the degree $g$. The results confirm our previous analysis: compared to "answer directly", prompting the LLM calls to "calculate step by step" significantly boosts the accuracy of the overall algorithm and leads to satisfactory performance for a wide range of task configurations, while incurring higher costs in the latency and total number of decoding tokens by a multiplicative factor, and making no difference in the number of LLM calls or total number of prefilling tokens.

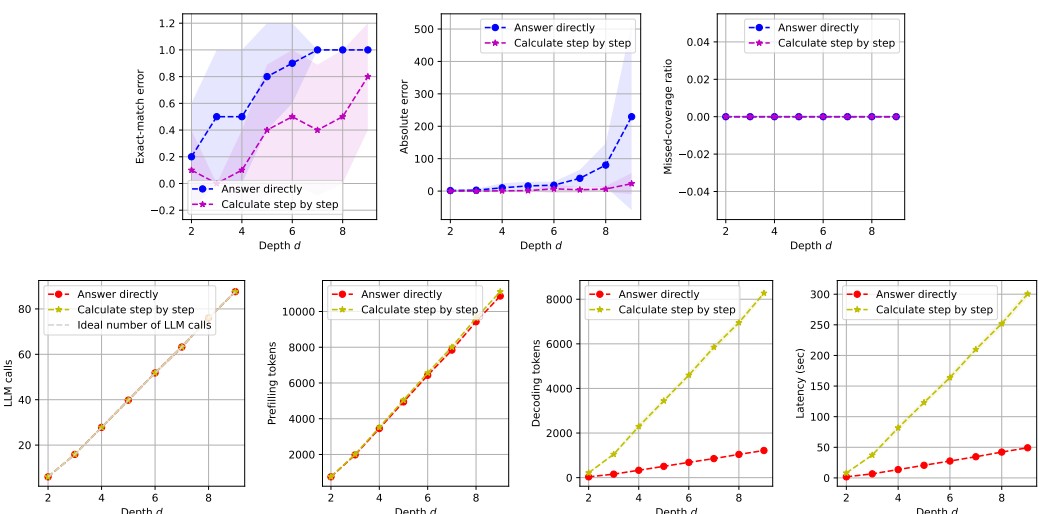

Figure 17: Empirical results (10 trials) for reasoning with recursive task decomposition, where two options of prompting the LLM calls responsible for calculation are compared. The width $w = 6$ and degree $g = 4$ are fixed, while the depth $d$ varies.

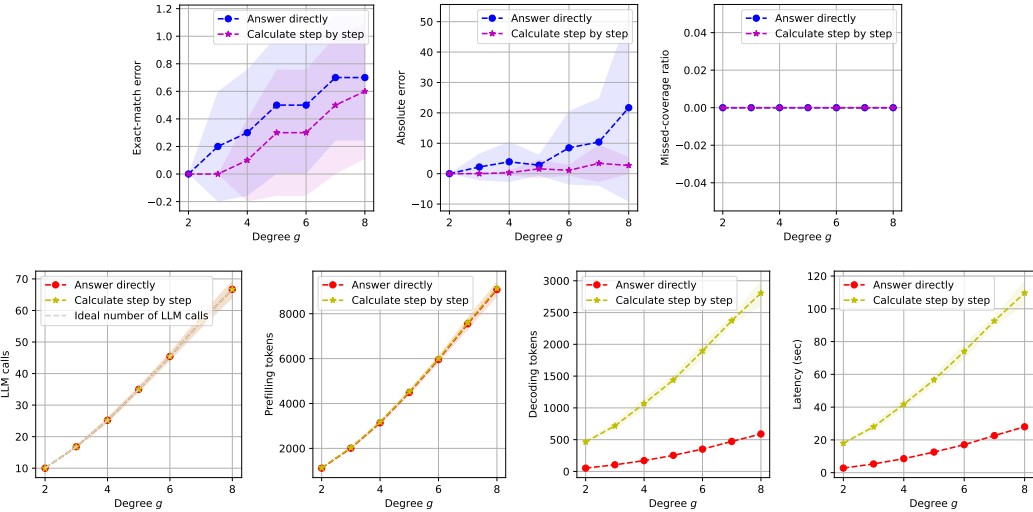

Figure 18: Empirical results (10 trials) for reasoning with recursive task decomposition, where two options of prompting the LLM calls responsible for calculation are compared. The depth $d = 3$ and width $w = 100$ are fixed, while the degree $g$ varies.

### E.5 Proof of Proposition 3

We prove the proposition by induction. First, consider the base case where $v$ is a leaf node of the DAG that has no input. Then it can be checked that Eq. (8) holds true:

$$\mathcal{E}(y(v)) \leq \mathcal{E}_v = \sum_{w \in \text{DAG}} \sum_{\text{path} \in \mathcal{P}(w \to v)} S^{|\text{path}|} \times \mathcal{E}_w.$$

The inequality follows the assumption, while the equality holds true because $\mathcal{P}(w \to v)$ for a leaf node $v$ is an empty set unless $w = v$, in which case $\mathcal{P}(w \to v)$ contains only a hypothetical path of length 0.

Next, consider a non-leaf node $v$, whose predecessor nodes are denoted by $u_1, u_2, \ldots, u_k$. Suppose that Eq. (8) holds true for each $u_i$, namely

$$\mathcal{E}(y(u_i)) \leq \sum_{w \in \text{DAG}} \sum_{\text{path} \in \mathcal{P}(w \to u_i)} S^{|\text{path}|} \times \mathcal{E}_w, \quad i \in [k].$$

To prove that Eq. (8) also holds true for node $v$, we start with the following:

$$\mathcal{E}(y(v)) \leq \mathcal{E}_v + S \times \sum_{i \in [k]} \mathcal{E}(y(u_i))$$

$$\leq \mathcal{E}_v + S \times \sum_{i \in [k]} \sum_{w \in \text{DAG}} \sum_{\text{path} \in \mathcal{P}(w \to u_i)} S^{|\text{path}|} \times \mathcal{E}_w$$

$$= \mathcal{E}_v + S \times \sum_{i \in [k]} \sum_{w \in \text{DAG} \setminus \{v\}} \sum_{\text{path} \in \mathcal{P}(w \to u_i)} S^{|\text{path}|} \times \mathcal{E}_w$$

$$= \mathcal{E}_v + S \times \sum_{w \in \text{DAG} \setminus \{v\}} \sum_{i \in [k]} \sum_{\text{path} \in \mathcal{P}(w \to u_i)} S^{|\text{path}|} \times \mathcal{E}_w$$

$$= \mathcal{E}_v + \sum_{w \in \text{DAG} \setminus \{v\}} \sum_{i \in [k]} \sum_{\text{path} \in \mathcal{P}(w \to u_i)} S^{|\text{path}|+1} \times \mathcal{E}_w.$$

Here in the third line, we replace the summation over $w \in \text{DAG}$ with $w \in \text{DAG} \setminus \{v\}$, since there is certainly no path from node $v$ to its predecessor node $u_i$. To move forward, notice that $\mathcal{P}(w \to v)$ is a union of $k$ disjoint sets, where the $i$-th set contains paths ending with the directed edge from $u_i$ to $v$:

$$\mathcal{P}(w \to v) = \bigcup_{i \in [k]} \left\{ \text{path} + [u_i \to v] : \text{path} \in \mathcal{P}(w \to u_i) \right\}.$$

With this, we can further simplify the previous upper bound:

$$\mathcal{E}(y(v)) \leq \mathcal{E}_v + \sum_{w \in \text{DAG} \setminus \{v\}} \sum_{i \in [k]} \sum_{\text{path} \in \mathcal{P}(w \to u_i)} S^{|\text{path}|+1} \times \mathcal{E}_w$$

$$= \mathcal{E}_v + \sum_{w \in \text{DAG} \setminus \{v\}} \sum_{\text{path} \in \mathcal{P}(w \to v)} S^{|\text{path}|} \times \mathcal{E}_w$$

$$= \sum_{w \in \text{DAG}} \sum_{\text{path} \in \mathcal{P}(w \to v)} S^{|\text{path}|} \times \mathcal{E}_w.$$

This concludes our proof of the proposition.

