# OpenReview forum: "On the Design and Analysis of LLM-Based Algorithms"
_ICLR.cc/2025/Conference — Submitted to ICLR 2025_

### Official Review · Reviewer_ZsEg · 2024-10-21

**Soundness:** 3
**Presentation:** 2
**Contribution:** 1
**Rating:** 3
**Confidence:** 3

**Summary:**

The aim of the paper is to analyze the behavior of LLM-based algorithms, specifically their theoretical "accuracy" and cost. The authors present LLM-based algorithms as computational graphs consisting of LLM queries and non-LLM computations that capture the data flow and allow for analysis via dynamic aggregation. The proposed framework is described in Section 2. Then, Sections 3.0-3.1 provide an abstract analysis of a map-reduce pattern, which is followed by special examples of counting and retrieval. In Section 4, a hierarchical decomposition pattern is analyzed on an example of determining a value of a variable. Finally, Section 5 analyses an example composed of recursive queries.

**Strengths:**

I like the proposed idea of analyzing the LLM-based algorithms as computational graphs. It's definitely the most natural way, similarly to analyzing standard algorithms and data flow.

**Weaknesses:**

The main problem of that paper is that, in my opinion, there is no takeaway from it. The proposed analysis mostly ends on framing the selected patterns as computational graphs, showing a basic bound, sometimes followed by a very generic statements like "so there is a tradeoff, period". I do think that we should have a principled way of analyzing LLM-based algorithms and the proposed framework looks promising, but little was done to use it. To be more precise:

- In Section 3 you analyze the map-reduce pattern, and in my opinion the most interesting things happen here.
    - You derive (or rather state since it is straightforward) the bound on the cost (Equation 4). Then, you use it to show (actually state) that it is minimized at $m=min(n, \bar m)$. Yes, I agree that if the cost is linear or sub-linear with respect to the input, the best idea is to use as large chunks as possible (again, rather a straightforward statement).
    - Then, you derive a bound on the cost with quadratic complexity and find a minimizer for that, which is approximately $L_{sys}$. This is actually interesting and surely non-trivial, but sadly no further discussion is provided.
    - Then, you analyze the parallel setting. Although the analysis is sound, I was a bit disappointed when I understood the key takeaway: "when m is very big, the cost increases if we make it even bigger; when m is very small, the costs increases if make it even smaller". Unfortunately, I find it straightforward -- there is always a tradeoff in the size of distributed computation parts and using too small or too big is never a good idea.
    - Then, you state that the cost is minimized by $m\asymp n/p$. That would be interesting, but I cannot find any justification for that statement.
    - Also, the asymptotic notation you use is a bit hard to parse. The only variable that has unbounded support is n, which should make other variables either constants or implicit functions of n, which is not specified directly. It looks like you take asymptotic of m, although it's bounded by n.
    - The "implication" in line 323 literally states that "The optimal value of m might depend on various factors, period".
    - Section 3.2 applies the derived formulas to a counting example. However, (1) the cost analysis is little beyond simply rewriting the formulas, (2) the takeaway is again straightforward (overall counting error is smaller if chunks are smaller)
    - Section 3.3 describes clearly the needle-in-a-haystack example. But then, equally clearly states the conclusion "the optimal value of m is a tradeoff, it can't be too big or too small period". Please, explain me why any kind of analysis is needed to make such a claim?

- In Section 4 you decribe the hierarchical decomposition pattern. Now, there are two listed conclusions: (1) reasoning has much lower cost than retrieval, (2) making the algorithm sequential or parallel has pros and cons. Although I agree with both, at the same time I don't see how did your framework help you derive those. They were not conclusions, you stated both during analysis and that's fine, since both need no explanation. But then, again, you haven't shown benefits of using your framework.

- In Section 5 you analyze recursive decomposition and state a bound on the error (and that's literally all). The question is: why do we need that bound? Is it helpful in any way? Why there is no discussion?

- As detailed, although I like the high-level idea of the framework, the paper shows no significant usage of it. Having said that, I'd be happy to be proven wrong, so I'm open to the discussion.

Other comments:

- I suggest adding a brief example of any LLM-based algorithm in the beginning. Initially I thought more about algorithms like Dijkstra, so specifying that with a simple one-sentence example would be helpful.
- It takes 4 pages until you start any analysis. I suggest making the initial descriptions (in fact, everything) much more concise, since the framework you propose is rather simple (which is a benefit), but then reading 4 pages of "how to decompose an algorithm into a computation graph" sounds much too lengthy.
- It's a bit confusing that you name the paragraphs the same way in sections 2.3 and 3.1. If you repeat the same names in Section 2 (which is introducing the framework), then in 3.1 it sounds like an unintended repetition at first glance.
- Figure 13 is actually very helpful in understanding the description. Please move it to the main text, even one of them, even smaller.
- The paper is missing a discussion of limitations and related works.

**Questions:**

Please list the non-trivial takeaways stemming from your analysis. How does it influence the design of LLM-guided algorithms? What does it explain in those we already have? How does it contribute to the field? Please be precise.

**Details Of Ethics Concerns:**

I have no concerns.

---

> ### Author Response · Authors · 2024-11-21
> **Author response to Reviewer ZsEg**
>
> Thank you for your careful reading and extensive comments. We hope our response can address your concerns.
>
>
> > Takeaways from this work?
>
> If there is a single takeaway from this work, it would be the proposed analytical framework described in Section 2.
> The primary goal of our work is to share this useful tool with anyone working on LLMs and LLM-based algorithms.
>
>
> The case studies in Sections 3, 4, 5 (and in the appendix) mainly serves as examples of how to use the proposed framework for solving concrete problems.
> If we draw an analogy between the proposed framework and PyTorch,
> then the case studies in our work are like the quickstart examples in PyTorch's official tutorials
> (e.g., "how to analyze a parallel-decomposition algorithm for the counting task using the analytical framework" is just like "how to build and train a convolutional neural network using PyTorch"),
> which help the users get started with using the proposed tool to solve their own problems.
>
>
> The case studies also serve the purpose of validating the usefulness of the proposed framework and showing what can be achieved with it.
> Of course, its usefulness for a particular reader of this work can vary,
> and it is understandable that some readers may find it unnecessary,
> just like one can always build and train a CNN from scratch using NumPy alone and without the need of PyTorch or any other deep learning frameworks.
>
>
> If there must be some takeaways from our analysis in the case studies,
> they would be the *quantitative and analytical* results that were non-trivially obtained with the proposed framework,
> rather than those wrap-up comments or casual remarks that you quoted in many of your comments and mistakenly regarded as the takeaways that we try to convey.
> Examples of such takeaways include the math expressions for the total cost with quadratic complexity and its minimizer in Lines 303 - 310 (which you quoted in one of your comments for Section 3),
> or the formula $m \asymp n / p$ in Line 321 that explains what we precisely mean when we say something like "the overall cost is minimized at some intermediate value of $m$".
>
>
> We believe that this response can address most of your comments about the takeaways from this work.
> Further responses are presented below to some of these comments, and skipped for the remaining ones to avoid repetition.
>
>
> > Then, you derive a bound on the cost with quadratic complexity and find a minimizer for that, which is approximately $L_{sys}$. This is actually interesting and surely non-trivial, but sadly no further discussion is provided.
>
> The math expressions for the cost and minimizer seem like the kind of "precise and non-trivial takeaways" that you are looking for,
> while a further discussion will likely fall into the category of "the optimal value of m might depend on various factors, period" comments that you strongly dislike.
> Could you provide some hints on what kind of further discussions would be useful here?
>
>
> > Then, you state that the cost is minimized by $m \asymp n / p$. That would be interesting, but I cannot find any justification for that statement.
>
> The justification can be found in Lines 315 - 321, right above the quoted statement.
>
>
> > The "implication" in line 323 literally states that "The optimal value of m might depend on various factors, period".
>
> The quoted line is just a wrap-up comment for the quantitative analysis above it, not a takeaway message that we try to convey.
>
>
> > Section 3.2 applies the derived formulas to a counting example. However, (1) the cost analysis is little beyond simply rewriting the formulas, (2) the takeaway is again straightforward (overall counting error is smaller if chunks are smaller)
>
>
> - We rewrite the formulas for the cost metrics, so that a reader does not have to go back and forth when reading this part.
>
> - The formulas from our analysis for parallel decomposition can be reused not only here but also in other concrete examples of parallel decomposition, or even in more complex algorithms that use parallel decomposition as a building block.
> Analyzing everything from scratch is thus unnecessary in these cases.
> This is exactly one of the benefits of having a unified analysis, isn't it?
>
>
> - We agree that the takeaway about the overall counting error looks straightforward,
> once the analysis in Lines 343 - 350 has been presented.
> After all, as mentioned in Line 333, the "counting" example is merely a warm-up exercise that demonstrates how to use the proposed framework in solving a concrete problem.

---

> ### Author Response · Authors · 2024-11-21
> **Author response to Reviewer ZsEg**
>
> > Comments on writing
>
> Thank you for the suggestions. We will incorporate them in the revision.
>
>
> > Asymptotic notation
>
> The confusion seems to arise because we use the notation $n$ in Lines 279 - 284 where we explain the generic asymptotic notation, which coincides with the notation $n$ in our analysis for parallel decomposition.
> We will use a different variable name in Lines 279 - 284 in the revision to fix this issue.
> In our analysis for parallel decomposition, $n$ is not the only variable that can grow to infinity.
>
>
> > The paper is missing a discussion of limitations and related works.
>
> As mentioned in Section 6, we defer the full discussion on related works to Appendix A, and discussion on limitations and future work to Appendix B, due to space limitations.
>
>
> > Reading 4 pages of "how to decompose an algorithm into a computation graph" sounds much too lengthy.
>
> Since the proposed analytical framework described in Section 2 is the core component of this work, we think it might be useful to make the descriptions as precise and detailed as possible.
> While readers who are proficient in the topic of this work might find it a bit lengthy, other readers with different backgrounds might feel that the length is just fine.
> Also, the content in the first 4 pages is not merely about "how to decompose an algorithm into a computation graph" like you described in your (somewhat sarcastic) comment.

---

> > ### Comment · Reviewer_ZsEg · 2024-11-23
> >
> > Thank you for your answers.
> >
> > > If there is a single takeaway from this work, it would be the proposed analytical framework described in Section 2. The primary goal of our work is to share this useful tool with anyone working on LLMs and LLM-based algorithms.
> >
> > As I stated before, the idea of the proposed framework does look interesting to me. However, a strong paper should not only introduce the framework, but also _convince_ people to use it.
> >
> > > The case studies also serve the purpose of validating the usefulness of the proposed framework and showing what can be achieved with it.
> >
> > And this is exactly what I mean by that. I believe the latter part is at least equally important as introducing the framework. Just like introducing PyTorch needs both the documentation AND examples of SOTA networks and trainings, to ground the messages "our framework is easier to use" and "our framework leads to better results". Otherwise, why to use it?
> >
> > > If there must be some takeaways from our analysis in the case studies, they would be the quantitative and analytical results that were non-trivially obtained with the proposed framework, rather than those wrap-up comments or casual remarks that you quoted in many of your comments and mistakenly regarded as the takeaways that we try to convey.
> >
> > First thing is that if I failed to identify the key takeaways from the analysis, it _may_ mean that they should be highlighted a bit more, or summarized at the end, or made more visible in any way. I can't argue whether specific results are non-trivial or not, since it's mostly a subjective opinion. However, one possible way of presenting an unquestionably non-trivial takeaway is to show "People widely use algorithm X. Empirically they use parameter Y. Our framework shows that the optimal parameter is Z, and indeed it works better in practice." Please note that it _may not_ require changing your analysis, but rather framing it in more clear and convincing way. Currently, it lacks that property -- as far as I see it is not only my opinion.
> >
> > ---
> >
> > > Could you provide some hints on what kind of further discussions would be useful here?
> >
> > No theory is for theory itself. At least one sentence that summarizes the takeaway would do. Maybe a simple "So to minimize a cost, it is usually beneficial to use m\approx L_sys". Or perhaps even better, you can identify cases in which other terms in that expression surprisingly start to dominate and that changes a lot, so e.g. "Usually m=L_sys, but if the environment has property X, then gamma dominates, hence we should use m=sqrt(gamma/alpha)". Or any other discussion that convinces the reader that this result is meaningful beyond simply stating it.
> >
> > > The quoted line is just a wrap-up comment for the quantitative analysis above it, not a takeaway message that we try to convey.
> >
> > I see. Then I suggest not starting the last paragraph from "One implication of the above analysis is".
> >
> > > we think it might be useful to make the descriptions as precise and detailed as possible.
> >
> > Please keep it concise. If you make it too detailed, it is not helpful for anyone.
> >
> > > Also, the content in the first 4 pages is not merely about "how to decompose an algorithm into a computation graph" like you described in your (somewhat sarcastic) comment.
> >
> > I apologize if you feel offended. My goal is to give you an honest feedback to make your paper stronger. That was exactly my impression, one you definitely want to avoid. You may take it into account or not.
> >
> > ---
> >
> > Overall, I think the paper has potential, but the presentation needs more work to become convincing and constitute a good contribution for the top conference. For now, I keep my score.

---

> ### Author Response · Authors · 2024-12-02
>
> Thank you for your reply. We do appreciate your thoughtful comments and suggestions, which will certainly be helpful in improving the writing and presentation of our work.

---

### Official Review · Reviewer_eLge · 2024-11-01

**Soundness:** 3
**Presentation:** 2
**Contribution:** 2
**Rating:** 3
**Confidence:** 4

**Summary:**

The paper proposes a framework to formally investigate LLM-based algorithms,
allowing to assess accuracy and efficiency. The authors describe their
framework and instantiate a few different LLM-based algorithms, which they then
analyze.

**Strengths:**

The paper tackles an interesting and important problem.

**Weaknesses:**

First, the paper is far too long for a conference format (the appendix is twice
as long as the main paper and includes sections that should be part of the main
paper, like related work). This paper would be more suitable as a journal paper.

With regards to the proposed evaluation framework, very little of it seems to be
specific to LLMs, or rather the LLM-specific parts are provided by the
investigator. It seems that this would be difficult in practice -- how would I
characterize the capabilities of any given LLM in a way that allows to determine
what the output would be for a given prompt?

The insights the analyses in the paper provide are very generic: "the optimal
value of m that minimizes costs might depend on the choices of cost metrics and
assumptions of LLM inference service, among other factors", "the minimum error
of the overall algorithm might be obtained by some intermediate value of m that
achieves a balance between these two failure modes", "each option of retrieval
has its own pros and cons". None of these are actionable, and it is unclear that
the proposed framework is necessary to obtain them. It is unclear whether other
insights are generally true, in particular "since a smaller value of m makes
each sub-task easier, it is reasonable to expect that the overall error E(y)
with this metric will also become smaller as m decreases" -- while the
individual errors might decrease, combining multiple steps potentially compounds
individual errors, resulting in an overall increase in error.

Other parts of the proposed framework are unclear. Section 3.3 describes the
answer being generated by majority voting, but the correct answer appears only
once. Dividing the input text into chunks, it seems that there would be a lot of
incorrect and "don't know" answers and, hopefully, a single correct one (for the
chunk that did contain the answer). How can majority voting possibly return the
correct result in this case?

**Questions:**

See weaknesses.

Update after responses: Thank you for your responses.

---

> ### Author Response · Authors · 2024-11-21
> **Author response to Reviewer eLge**
>
> Thank you for your review. We hope our response can address your concerns.
>
> > The insights the analyses in the paper provide are very generic: "the optimal value of m that minimizes costs might depend on the choices of cost metrics and assumptions of LLM inference service, among other factors", "the minimum error of the overall algorithm might be obtained by some intermediate value of m that achieves a balance between these two failure modes", "each option of retrieval has its own pros and cons". None of these are actionable, and it is unclear that the proposed framework is necessary to obtain them.
>
> Each of the statements that you quoted has been accompanied by concrete, precise and non-trivial analyses obtained with the proposed framework:
> Lines 286 - 321 for the first quote, Lines 386 - 400 for the second, and Lines 458 - 460 for the third.
> Each of the quoted statements is merely a wrap-up comment for the corresponding analyses right above it,
> rather than an insight or key message that we try to convey.
>
>
> > It is unclear whether other insights are generally true, in particular "since a smaller value of m makes each sub-task easier, it is reasonable to expect that the overall error E(y) with this metric will also become smaller as m decreases" -- while the individual errors might decrease, combining multiple steps potentially compounds individual errors, resulting in an overall increase in error.
>
> - We made the quoted claim merely for the "counting" example, and the claim has been backed by both formal analysis (Lines 347 - 349) and empirical validation (Appendix C.1).
>
> - We did not claim that the quoted statement must be generally true and, like you said, there are of course cases where it is not.
>
> - Other examples where the overall error of an LLM-based algorithm depends *gracefully* on the individual errors include the "sorting" example (Proposition 2 in Appendix C.2.2), as well as the case of additive errors and bounded sensitivity (Proposition 1 with sensitivity parameter $S < 1$). The quoted insight for each of these examples is again backed by formal analysis (thanks to the proposed analytical framework) and empirical validation.
>
>
> > Section 3.3 describes the answer being generated by majority voting, but the correct answer appears only once. Dividing the input text into chunks, it seems that there would be a lot of incorrect and "don't know" answers and, hopefully, a single correct one (for the chunk that did contain the answer). How can majority voting possibly return the correct result in this case?
>
> It is reasonable to use majority voting because we adopt chunking *with overlaps* in this task, as mentioned in Appendix C.3.1;
> we will add this note to Section 3.3 in the revision.
> In this case, the correct answer can appears in multiple chunks, and thus majority voting (with the "don't know" answers excluded, as noted in Line 382) can be helpful when incorrecet answers appear occasionally.
> More details about majority voting in this example can be found in Appendix C.3.1.

---

> ### Author Response · Authors · 2024-11-21
> **Author response to Reviewer eLge**
>
> > How would I characterize the capabilities of any given LLM in a way that allows to determine what the output would be for a given prompt?
>
> It is true that determining what the exact output by the LLM would be for a given prompt is infeasible,
> but that should not prevent us from trying to understand and analyze the performance of LLM-based algorithms.
> Our analytical framework is proposed in an effort to achieve this,
> and we have shown throughout this work that meaningful analysis is possible without the need for exact characterization of LLMs.
>
> For example, by applying the analytical framework to the "counting" task described in Section 3.2, we are able to predict how the performance of the LLM-based algorithm are impacted by the hyperparameter $m$,
> based on two weak and natural assumptions about the LLM:
> (1) the error in solving a sub-task of size $m$ is increasing in $m$, and
> (2) for each sub-task, the LLM generates a response of length $\mathcal{L}_{\text{dec}} = O(1)$.
>
> In practice, one can leverage measuring and profiling in order to better characterize the capabilities of LLMs.
> A related discussion on these practical considerations can be found in Appendix B.2.
>
>
> > The paper is far too long for a conference format (the appendix is twice as long as the main paper and includes sections that should be part of the main paper, like related work). This paper would be more suitable as a journal paper.
>
> - There have been many ICLR/ICML/NeurIPS papers that have similar length,
> and ICLR allows an appendix of unlimited length.
> Similarly, many papers in the past defer full discussion on related works to the appendix, like we do in this manuscript.
>
> - Most of the appendix in our manuscript is about case studies and concrete examples,
> which are non-essential to our core contribution, namely the proposed analytical framework.
> A reader of this work should feel free to refer to the appendix selectively, based on their own needs.

---

### Official Review · Reviewer_gk9V · 2024-11-03

**Soundness:** 2
**Presentation:** 3
**Contribution:** 1
**Rating:** 3
**Confidence:** 2

**Summary:**

The paper attempts to formalize cost and accuracy analysis of LLM-based algorithms .

**Strengths:**

The paper advocated analysis of LLM-based algorithms.

**Weaknesses:**

The theoretical part proposes a framework for analysis which looks like a simplistic variant of analysis of parallel algorithms, something one learns during undergrad CS studies. The 'empirical' evaluation in the body of the paper is just a qualitative description of application of the proposed (rather standard) methodology to a few problems. There is 'numerical evaluation' in the appendix, which is not convincing and lacks detail.

While I welcome the idea of systematic analysis of algorithms, including LLM-based ones, the paper lacks both theoretical novelty and empirical justification. Significant effort has to be spent to bring this paper to the level of a publication at a major conference such as ICLR.

**Questions:**

What part of your analysis is applicable exclusively to LLMs rather than to any parallel algorithm using external resource?

---

> ### Author Response · Authors · 2024-11-21
> **Author response to Reviewer gk9V**
>
> Thank you for your review. We hope our response can address your concerns.
>
> > The theoretical part proposes a framework for analysis which looks like a simplistic variant of analysis of parallel algorithms, something one learns during undergrad CS studies.
>
> This claim seems to miss a large part of the content of our work.
> In fact, the analysis for parallel decomposition (which has its own complications due to the use of LLMs, compared to analysis of parallel algorithms in other domains) only occupies a small fraction of our analysis in this work.
>
>
> > The 'empirical' evaluation in the body of the paper is just a qualitative description of application of the proposed (rather standard) methodology to a few problems.
>
> Could you clarify which "empirical evaluation in the body of the paper" are you referring to?
> As far as we can tell, all empirical/numerical evaluation has been deferred to the appendix.
>
>
> > "There is 'numerical evaluation' in the appendix, which is not convincing and lacks detail."
> > "While I welcome the idea of systematic analysis of algorithms, including LLM-based ones, the paper lacks both theoretical novelty and empirical justification. Significant effort has to be spent to bring this paper to the level of a publication at a major conference such as ICLR."
>
> We would appreciate more concrete and actionable feedback on our work,
> e.g.,
> why the numerical evaluation is not convincing, what additional details will be useful,
> references to prior works that make it plausible to deem our work to be "lacking theoretical novelty",
> etc.
>
>
> > What part of your analysis is applicable exclusively to LLMs rather than to any parallel algorithm using external resource?
>
> Examples for this include:
>
> - We incorporate the diversity of LLM inference service in our analysis of efficiency and cost metrics.
> For example, the various forms of the cost function $C_{LLM}(L_{pre}, L_{dec})$ are specific to LLMs.
>
> - We incorporate the inexactness of LLM's computation in our analysis of accuracy and error metrics.
> For example, in parallel decomposition, the choice of the sub-task size $m$ impacts not only efficiency but also accuracy of the overall algorithm, which is not the case for parallel algorithms in many other domains (e.g., DNN training with data parallelism).
>
> - Any part of our analysis that is not about parallel algorithms.

---

> > ### Comment · Reviewer_gk9V · 2024-11-21
> > **empirical evaluation**
> >
> > The abstract states
> >
> > "Through extensive analytical and empirical
> > investigation in a series of case studies, we demonstrate that the proposed
> > framework is broadly applicable to a wide range of scenarios and diverse
> > patterns of LLM-based algorithms, such as parallel, hierarchical and recursive
> > task decomposition."
> >
> > The body of the paper, contrary to that, does not include empirical evaluation. Appendices are nice to have if questions arise, but they are not a part of the paper.

---

> > ### Comment · Reviewer_gk9V · 2024-11-21
> > **actionable feedback**
> >
> > I do not see novelty or significant contribution in this submission, whether in its current form or in a revision. This may probably be a position paper, if written and submitted appropriately. To contribute to LLM research, I'd suggest focusing on a concrete problem and solving it. If an analysis methodology emerges from that, then great.

---

### Official Review · Reviewer_VNRn · 2024-11-03

**Soundness:** 2
**Presentation:** 2
**Contribution:** 2
**Rating:** 3
**Confidence:** 4

**Summary:**

This paper is about a formal investigation into the design and analysis of LLM-based algorithms, i.e. algorithms that contain one or multiple calls of large language models (LLMs) as sub-routines and critically rely on the capabilities of LLMs

**Strengths:**

The stated objectives of the paper are excellent if they can be achieved.

**Weaknesses:**

The paper does not seem to deliver very much of the stated contributions. Proposition 1 is the key result, and it is quite weak.

The real issue is that a computation graph is a precise mathematical object that uniquely defines a computation, whereas this LLM-based computation graph is neither precise nor does it uniquely define a computation. The lack of uniqueness stems from the non-determinism of any LLM-based call.
As a consequence, how can one compare a "normal" computation graph with this LLM-based computation graph?
Further, the analysis should be stochastic (with expected complexity), rather than the proposed deterministic complexity. An LLM is inherently stochastic, so one cannot specify the outcome of an LLM call as a deterministic object.

The paper states: use “accuracy” to refer to the broader concept of “quality”, and an “error metric” can be any metric that measures how much the output of an algorithm deviates from certain criteria."

Where do the costs come from in  C(prefilling), C(decoding)?

It seems like all of these costs are just qualitative.

hypothetically categorize LLMs into two types: Type-1 LLMs are only prone to the first failure mode, while Type-2 LLMs are prone to both

Too much speculation in this article

It seems like the article builds up to Proposition 1,and then we ask "so what!". This is a pretty weak conclusion, and it does not even look like to has much strength as a formal mathematical expression.

**Questions:**

Please clarify if  all of the defined costs are just qualitative.

If they are not, how does one define a metric to assign values and to update the values?

---

> ### Author Response · Authors · 2024-11-21
> **Author response to Reviewer VNRn**
>
> Thank you for your review. We hope our response can address your concerns.
>
>
> > "Proposition 1 is the key result, and it is quite weak." "It seems like the article builds up to Proposition 1."
>
> This is a misunderstanding. Proposition 1 is not the key result of this work, and it is not even mentioned in the abstract or introduction section.
> The most important key result of this work should be the proposed analytical framework itself.
>
>
> > The paper states: use “accuracy” to refer to the broader concept of “quality”, and an “error metric” can be any metric that measures how much the output of an algorithm deviates from certain criteria."
>
> Could you clarify what is your concern about this statement?
>
>
> > Concerns about computational graphs, non-determinism, lack of uniqueness, stochastic analysis, etc.
>
> In fact, an LLM call can be deterministic, e.g., when greedy decoding is adopted.
> Even if stochastic decoding is adopted (along with other randomness), our analysis is conditioned on fixed random seeds (as noted in Lines 213 and 219), which again makes everything deterministic.
>
> Stochastic analysis for LLM-based algorithms is certainly a valuable direction for future work, and it can be built upon our deterministic analysis, though this is beyond the scope of our current work.
> A related discussion can be found in Lines 883 - 890.
>
>
> > Are the defined costs just qualitative?
>
> Our analysis of cost metrics is obviously *quantitative* rather than qualitative,
> as exemplified in Lines 228 - 246, Lines 286 - 323, Lines 435 - 452, among many others in the appendix.
>
>
>
> > "Hypothetically categorize LLMs into two types: Type-1 LLMs are only prone to the first failure mode, while Type-2 LLMs are prone to both". "Too much speculation in this article."
>
>
> - We make this simplified categorization merely for the purpose of explanation and illustration.
> And it is not just speculation, but actually backed by empirical observations (as discussed in Appendix C.3.3).
>
> - It feels unfair to say that there is "too much speculation in this article" merely based on this single sentence from one specific case study in our work.

---

### Meta-Review · Area_Chair_AWxc · 2024-12-07

**Metareview:**

The paper proposes a framework for analyzing the accuracy and efficiency of LLM-based algorithms.

The strength of the paper is the importance of the problem it is trying to address.

The main weakness of the paper is that the proposed framework is too straightforward to be considered a sufficient contribution. In other words, if one wanted to analyze, e.g., the cost of an LLM-based algorithm, it's not clear which part of the paper's analysis would be non-trivial for that person to derive on their own. This is especially true for the accuracy analysis because, as the authors admit in the paper, the concept of accuracy is task-specific, so much or all of its analysis is bespoke for each task.

Since this weakness is major and fundamental, the metareviewer recommends rejection.

**Additional Comments On Reviewer Discussion:**

The major points raised by the reviewers revolved around the weakness described above and their impression that hardly any of the analysis is specific to LLM-based algorithms (as opposed to being applicable to oracle-based algorithms in general). While Reviewer ZsEg found the framework to have some potential despite its conclusions seeming straightforward to this reviewer, the other reviewers found it difficult to identify the paper's strengths at all. Judging by the review contents, all reviewers were explicitly or implicitly confused by the paper's explanations.

The revised paper version and the authors' responses fixed some of the clarity issues but didn't convince the metareviewer and the reviewers that the contribution is non-straightforward enough to merit acceptance. This applies, among other things, to the authors' claim in the response to Reviewer gk9V that the cost functions are specific to LLM analysis: while this is true, deriving these cost functions is very straightforward, and the implications of the functions' structure are very straightforward as well.

Thus, the reviewers stand by their unanimous recommendation to reject the paper even after the rebuttal, and the metarevewer, who has read the paper as well, concurs.

---

### Decision · Program_Chairs · 2025-01-22

Reject